# Acute inhibition of centriolar satellite function and positioning reveals their functions at the primary cilium

Özge Z. Aydin[ö], Sevket Onur Taflan[iD][ö], Can Gurkaslar[iD], Elif Nur Firat-Karalar[iD]*

Department of Molecular Biology and Genetics, Koc University, Istanbul, Turkey

[ö] These authors contributed equally to this work.
* ekaralar@ku.edu.tr

**Data Availability Statement:** All relevant data are within the paper and its Supporting Information files.

## Abstract

Centriolar satellites are dynamic, membraneless granules composed of over 200 proteins. They store, modify, and traffic centrosome and primary cilium proteins, and help to regulate both the biogenesis and some functions of centrosomes and cilium. In most cell types, satellites cluster around the perinuclear centrosome, but their integrity and cellular distribution are dynamically remodeled in response to different stimuli, such as cell cycle cues. Dissecting the specific and temporal functions and mechanisms of satellites and how these are influenced by their cellular positioning and dynamics has been challenging using genetic approaches, particularly in ciliated and proliferating cells. To address this, we developed a chemical-based trafficking assay to rapidly and efficiently redistribute satellites to either the cell periphery or center, and fuse them into stable clusters in a temporally controlled way. Induced satellite clustering at either the periphery or center resulted in antagonistic changes in the pericentrosomal levels of a subset of proteins, revealing a direct and selective role for their positioning in protein targeting and sequestration. Systematic analysis of the interactome of peripheral satellite clusters revealed enrichment of proteins implicated in cilium biogenesis and mitosis. Importantly, induction of peripheral satellite targeting in ciliated cells revealed a function for satellites not just for efficient cilium assembly but also in the maintenance of steady-state cilia and in cilia disassembly by regulating the structural integrity of the ciliary axoneme. Finally, perturbing satellite distribution and dynamics inhibited their mitotic dissolution, and mitotic progression was perturbed only in cells with centrosomal satellite clustering. Collectively, our results for the first time showed a direct link between satellite functions and their pericentrosomal clustering, suggested new mechanisms underlying satellite functions during cilium assembly, and provided a new tool for probing temporal satellite functions in different contexts

## Introduction

The mammalian centrosome/cilium complex consists of the centrosome, the cilium, and the centriolar satellites, which together regulate polarity, signaling, proliferation, and motility in

**Funding:** This work was supported by European Research Council Starting Grant 679140 to ENF-K (https://erc.europa.eu/starting-grants), Royal Society Newton Advanced Fellowship NA160060 to ENF-K (https://royalsociety.org/grants-schemes-awards/grants/newton-advanced-fellowships/), European Molecular Biology Organization Installation Grant 3622 to ENF-K (https://www.embo.org/funding-awards/installation-grants), and European Molecular Biology Organization Young Investigator Award to ENF-K (https://www.embo.org/funding-awards/young-investigators). The funders had no role in study design, data collection and analysis, decision to publish, or preparation of the manuscript.

**Competing interests:** The authors have declared that no competing interests exist.

**Abbreviations:** AGC, automatic gain control; Arl13b, ADP ribosylation factor-like GTPases 13b; BFDR, Bayesian false discovery rate; BICD2, bicaudal D homolog 2; BioID, Biotin Identification; C-Nap1, centrosomal Nek2-associated protein 1; CCDC14, coiled-coil domain containing 14; CCDC66, coiled-coil domain containing 66; CCP5, cytosolic carboxypeptidase 5; CDK1, cyclin-dependent kinase 1; CSPP1, centrosome and spindle pole associated protein 1; DDA, data-dependent acquisition; DMEM, Dulbecco's Modified Eagle Medium; DYRK3, dual specificity tyrosine phosphorylation-regulated kinase 3; FBS, fetal bovine serum; FKBP, FK506 binding protein 12; FRB, FKBP12-rapamycin-binding; GFP, green fluorescent protein; GO, Gene Ontology; HA, hemagglutinin; HCD, higher-energy collisional dissociation; HDAC6, histone deacetylase 6; IFT20, intraflagellar transport 20; IFT74, intraflagellar transport 74; IMCD3, inner medullary collecting duct 3; Kif5b, kinesin family member 5B; KO, knockout; MAP4, microtubule-associated protein 4; MIB1, mindbomb E3 ubiquitin ligase 1; MTOC, microtubule organizing center; mTOR, mammalian target of rapamycin; NCE, normalized collision energy; NUMA, nuclear mitotic apparatus; OFD1, oral-facial-digital syndrome 1; PCM1, pericentriolar material 1; Plk1, polo-like kinase 1; RPE1, retinal pigmented epithelial; SAINT, significance analysis of interactome; SIR, silicone rhodamine.

cells and thereby development and homeostasis in organisms. Centriolar satellites (hereafter satellites) are 70–100-nm membraneless electron-dense granules that localize and move around centrosomes and cilia [1]. Satellite assembly is scaffolded by the large coiled-coil protein pericentriolar material 1 (PCM1), which physically interacts with many known and putative centrosome proteins [2,3]. Satellite resident proteins function in a wide range of cellular processes, including cilium assembly, ciliary transport, centriole duplication, mitotic regulation, and microtubule dynamics and organization, and include proteins mutated in developmental and neuronal disorders [4,5]. Accordingly, acute or constitutive loss of satellites through depletion or deletion of PCM1 resulted in defects in cilium assembly, ciliary signaling, epithelial cell organization, and autophagy [5–8]. Satellites mediate their functions in part by regulating the cellular abundance or centrosomal and ciliary targeting of specific proteins [6–9]. These results, together with their microtubule-mediated active transport, led to the current model, which defines satellites as trafficking modules and/or storage sites for their protein residents.

Although satellites are ubiquitous structures in vertebrate cells, their number and distribution changes in a context-dependent way. For example, in most cell types, satellites are predominantly clustered around the centrosome, and to a lesser extent scattered throughout the cytoplasm [10]. However, in specialized cell types, their distribution varies from clustering at the nuclear envelope in myotubes and at the apical side of polarized epithelial cells, to being scattered throughout the cell body in neurons [10–12]. In addition to their cell type–specific distribution, satellite properties can also be influenced by extracellular stimuli. For example, genotoxic stress causes satellites to disperse throughout the cytoplasm, and during mitosis, satellites dissolve and subsequently reassemble upon mitotic exit [13,14]. The mitotic dissolution of PCM1 is regulated by a dual specificity tyrosine phosphorylation-regulated kinase 3 (DYRK3)-dependent mechanism and is a line of evidence for liquid-like behavior of satellites [13]. Consistent with such behavior, satellite granules were shown to undergo fusion and fission events with each other and the centrosome in epithelial cells [15]. These context-dependent variations in satellite properties suggest that their cellular distribution and remodeling have important implications for their functions, although the mechanisms remain unknown.

There are two key unknowns that pertain to our understanding of satellite functions as discrete protein complexes. First, despite its highly regulate nature, whether and, if so, how satellite distribution contributes to their functions is not known. Second, previous approaches that used depletion or deletion of PCM1 to study satellite functions were limited in uncovering their temporal functions, such as the ones during cilium maintenance and mitosis. Addressing these questions is an essential step in defining their relationship with centrosomes and cilia, and thereby the inter-organelle communication within the vertebrate centrosome/cilium complex. To this end, we used chemical manipulation to efficiently and rapidly target satellites to the microtubule plus ends at the cell periphery or microtubule minus ends at the cell center. Satellites targeted to these locations formed stable assemblies, and we showed that satellite proximity to centrosomes and cilia is required for their functions in centrosomal protein targeting, mitotic progression and cilium assembly, maintenance, and disassembly.

## Results

### Development and validation of the inducible satellite trafficking assay

Satellites cluster around the centrosome in most cell types, suggesting that their proximity to the centrosomes is important for their functions. To uncover temporal functions of satellites and to elucidate how their pericentrosomal clustering contributes to their functions, we developed an inducible satellite trafficking assay to target satellites away from the centrosomes to the cell periphery and determined the molecular and cellular consequences (Fig 1A). To

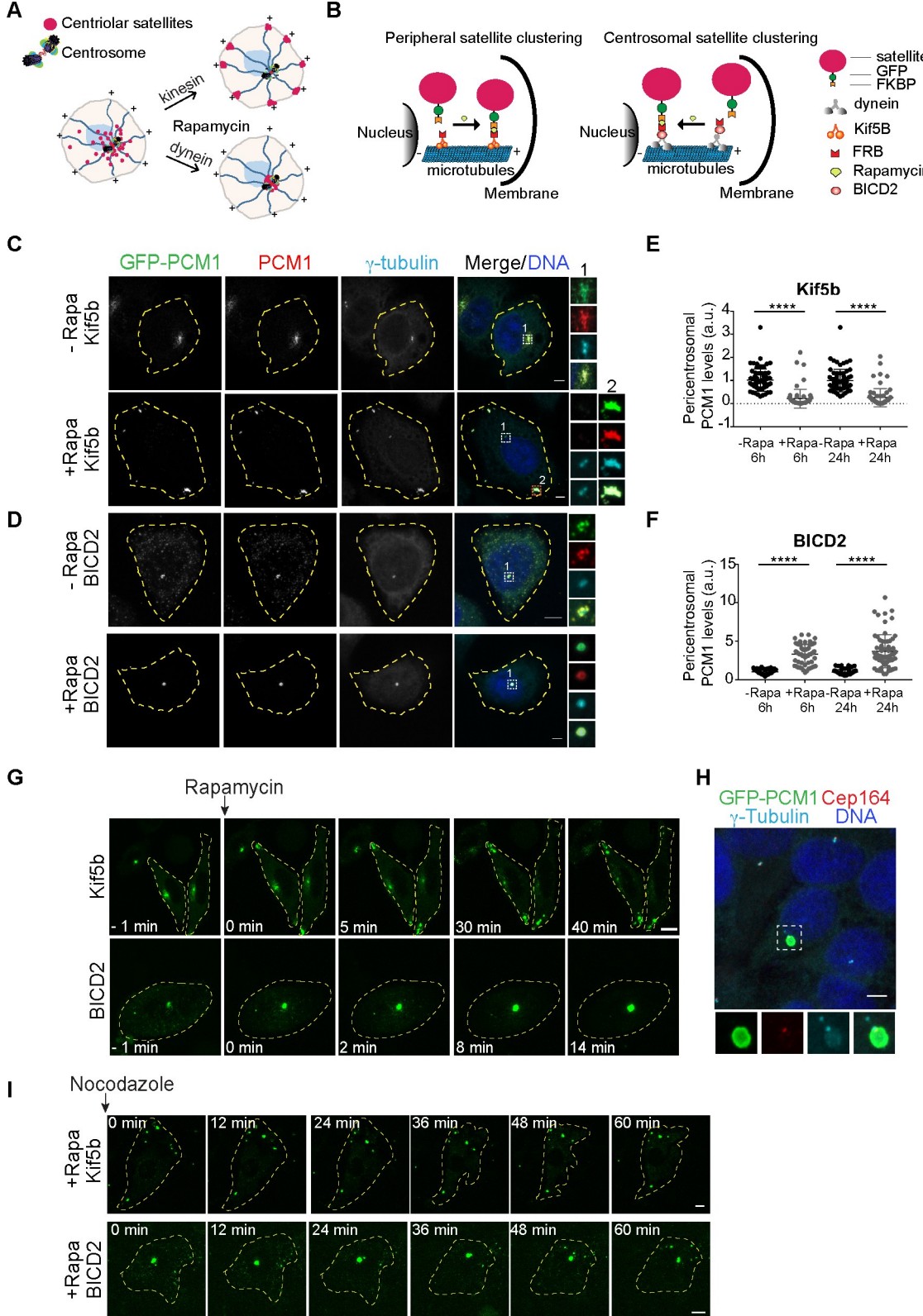

**Fig 1. Inducible dimerization of PCM1 with constitutively active motor domains targets satellites to the cell periphery or center.** Development and validation of the inducible satellite trafficking assay. (**A**) Representation of satellite redistribution to cell periphery or center upon inducible dimerization of the satellite scaffolding protein PCM1 with the constitutively active plus and

minus end–directed molecular motor proteins. **(B)** Design of the inducible satellite-trafficking assay. PCM1 was tagged with GFP at the N terminus and FKBP at the C terminus. Co-expression of GFP-PCM1-FKBP with HA-Kif5b (1–269 aa)-FRB and subsequent rapamycin induction targets satellites to the cell periphery, where microtubule plus ends are concentrated. Co-expression of GFP-PCM1-FKBP with HA-BICD2 (1–198 aa)-FRB and subsequent rapamycin induction targets satellites to the cell center, where microtubule minus ends are concentrated. **(C, D)** Representative images for satellite distribution in control and rapamycin-treated HeLa cells. Cells co-expressing GFP-PCM1-FKBP with HA-Kif5b-FRB or HA-BICD2-FRB were treated with rapamycin for 1 hour. Twenty-four hours after rapamycin induction, cells were fixed and stained with antibodies against GFP, PCM1, and gamma-tubulin. DNA was stained with DAPI. Cell edges are outlined. Scale bar, 5 μm. **(E, F)** Endogenous PCM1 exhibits the same localization pattern as GFP-PCM1-FKBP. Pericentrosomal PCM1 levels was quantified at 6 hours and 24 hours after rapamycin treatment of cells expressing GFP-PCM1-FKBP with **(E)** HA-Kif5b-FRB or **(F)** HA-BICD2-FRB. Average mean intensity of pericentrosomal levels in control cells were normalized to 1. $n \geq 25$ cells per experiment. Data represent mean value from two experiments per condition, ±SD (****$p < 0.0001$). **(G)** Dynamics of satellite redistribution to the cell periphery or center upon rapamycin addition. Cells co-expressing GFP-PCM1-FKBP with HA-Kif5b-FRB or HA-BICD2-FRB were imaged before and after rapamycin addition using time-lapse microscopy. Imaging was performed more frequently (every 2 minutes) in BICD2-expressing cells relative to Kif5b-expressing cells (every 5 minutes). Representative still frames from time-lapse experiments were shown. Cell edges are outlined. Time, minutes after rapamycin addition. Scale bar, 5 μm. **(H)** Satellites concentrate around mother centriole in cells co-expressing GFP-PCM1-FKBP with HA-BICD2-FRB after rapamycin treatment. Cells were stained for GFP, mother centriole marker Cep164, and gamma-tubulin. DNA was stained with DAPI. Cell edges are outlined. Scale bar, 5 μm. **(I)** Peripheral and centrosomal satellite clusters were maintained upon microtubule depolymerization. Cells with peripheral or centrosomal satellite clusters were treated with nocodazole for 1 hour and imaged every 3 minutes by time-lapse microscopy. Representative still frames from time-lapse experiments were shown. Cell edges are outlined. Scale bar, 5 μm. Error bars = SD. Source data can be found in S1 Data. BICD2, bicaudal D homolog 2; FKBP, FK506 binding protein 12; FRB, FKBP12-rapamycin-binding; GFP, green fluorescent protein; HA, hemagglutinin; Kif5b, kinesin family member 5B; PCM1, pericentriolar material 1; Rapa, rapamycin.

correlate the associated phenotypes with satellite proximity to the centrosomes, we in parallel targeted satellites to the cell center (Fig 1A). Our approach makes use of the inducible dimerization of the FK506 binding protein 12 (FKBP) and FKBP12-rapamycin-binding (FRB) domain of mammalian target of rapamycin (mTOR) by rapamycin or its cell-permeable analog, rapalog [16,17]. Due to the long half-life of rapamycin interaction with FKBP and FRB (about 17.5 hours), the dimerization is essentially irreversible [18]. This FKBP-FRB-based rapamycin-inducible heterodimerization approach was previously used to spatiotemporally manipulate specific cellular structures and catalytic activities such as the removal of intraflagellar transport 20 (IFT20) and intraflagellar transport 74 (IFT74) from cilia and sequestration at the mitochondria [19], ciliary targeting of cytosolic carboxypeptidase 5 (CCP5) tubulin deglutamylase [20], and recruitment of molecular motors to endosomes, peroxisomes, and the nucleus [21–24].

We adapted this assay for satellites by co-expressing FKBP fusion of satellites with the FRB fusion of constitutively active plus end or minus end–directed motor domains (Fig 1B). We tagged the satellite-scaffolding protein PCM1 with green fluorescent protein (GFP) at the N terminus for visualization of satellites and with FKBP at the C terminus for inducible dimerization with the motor domains (Fig 1B). Analogous to GFP-PCM1 and endogenous PCM1, GFP-PCM1-FKBP localized to satellites in transfected human cervical carcinoma (HeLa) cells without altering their distribution (S1A Fig). To target satellites to the microtubule plus ends at the cell periphery, we induced their recruitment to the FRB fusion of constitutively active kinesin-1 hemagglutinin (HA)–kinesin family member 5B (Kif5b) (1–269 aa) motor domain (hereafter, HA-Kif5b-FRB or Kif5b), which lacks the tail region required for interaction with endogenous cargoes and thus can only bind to satellites via the FKBP-FRB dimerization (Fig 1B) [23]. HA-Kif5b-FRB localized to the cytosol and did not affect the distribution of satellites (S1B Fig). To target satellites to the microtubule minus ends at the cell center, we induced their recruitment to the FRB fusion of the constitutively active N terminus of dynein/dynactin cargo adaptor bicaudal D homolog 2 (BICD2; 1–198 aa) (hereafter HA-BICD2-FRB or BICD2), which interacts with dynein and dynactin but lacks the cargo-binding domain (Fig 1B) [23]. HA-BICD2-FRB localized diffusely throughout the cytosol in low-expressing cells and formed granules in high-expressing cells (S1B Fig). In contrast to HA-Kif5b, we noticed

that HA-BICD2 expression by itself resulted in satellite dispersal throughout the cytosol (S1B Fig), likely through outcompeting endogenous activating adaptors that couple satellites to the dynein-dynactin complex. Of note, this is consistent with its previous characterization as a dominant-negative mutant that impairs dynein-dynactin function [25–27]. Although the activating adaptors required for minus end–directed movement of satellites are not known, our results validate that chemical dimerization of BICD2 with satellites is sufficient to induce their centrosomal clustering (Fig 1D, S1D Fig).

To test the feasibility of the inducible satellite trafficking assay, we first determined the localization of satellites in transfected HeLa cells before and after rapamycin addition by live imaging and quantitative immunofluorescence. In cells co-expressing GFP-PCM1-FKBP and HA-Kif5b-FRB, satellites had their typical pericentrosomal clustering pattern in the absence of rapamycin (Fig 1C, S1C Fig). Satellite localization ranged from clustering around the centrosomes to scattering throughout the cytosol in cells co-expressing GFP-PCM1-FKBP and HA-BICD2, depending on ectopic BICD2 expression levels (Fig 1D, S1D Fig). Upon rapamycin addition to cells expressing HA-Kif5b-FRB and GFP-PCM1-FKBP, satellites were targeted to the cell periphery, where the majority of microtubule plus ends localize (Fig 1C and 1G, S1C Fig, S1 Movie). In contrast, in cells expressing HA-BICD2-FRB and GFP-PCM1-FKBP, rapamycin addition resulted in clustering of satellites at the centrosome (Fig 1D and 1G, S1D Fig, S2 Movie). Notably, we observed partial distribution of both GFP-PCM1-FKBP and PCM1 to the cell periphery or center upon rapamycin addition in a subset of cells, likely due to insufficient expression of the active motors relative to the satellites (S1E Fig). The integrity and organization of the microtubule network remained unaltered in both cases upon rapamycin-induced redistribution of satellites (S1F Fig).

Importantly, localization of endogenous PCM1 tightly correlated with the localization of GFP-PCM1-FKBP before and after rapamycin addition (Fig 1C and 1D), which is in agreement with PCM1 self-dimerization through its N-terminal coiled-coil domains [10]. To determine the targeting efficiency of endogenous PCM1, we quantified its pericentrosomal levels by measuring fluorescence signal intensities within a 3-$\mu m^2$ circle encompassing the centrosome. We found that pericentrosomal levels decreased significantly in Kif5b-expressing (peripherally targeted) cells (6 hours: Control: $1 \pm 0.5$, Rapa: $0.3 \pm 0.5$, $p < 0.0001$; 24 hours: Control: $1.1 \pm 0.5$, Rapa: $0.3 \pm 0.4$, $p < 0.0001$) and increased significantly in BICD2-expressing (centrosomally targeted) cells (6 hours: Control: $1 \pm 0.4$, Rapa: $3.5 \pm 1.1$, $p < 0.0001$; 24 hours: Control: $1 \pm 0.5$, Rapa: $4.8 \pm 1.9$, $p < 0.0001$) at 6 hours and 24 hours after rapamycin induction (Fig 1E and 1F). As controls, rapamycin treatment of control cells or cells expressing only GFP-PCM1-FKBP did not perturb satellite distribution in cells, as assessed by staining cells with anti-PCM1 antibody (S1G Fig). Together, these results validated rapid and efficient redistribution of satellites to the cell periphery or center upon rapamycin-induced dimerization.

In Kif5b-expressing cells, satellites formed large clusters that were heterogeneously distributed at the peripheral cell protrusions, whereas in BICD2-expressing cells, they formed a single cluster at the centrosome (Fig 1C and 1D). A similar heterogeneous distribution pattern at the periphery was also observed when GFP-PCM1-FKBP was co-expressed with the motor domain of another kinesin, Kif17 (S1H Fig). Staining of BICD2-expressing cells for satellites and the mother centriole protein Cep164 revealed tight clustering of satellites around the mother centriole (Fig 1H). Given that a subset of centrosomal microtubules are anchored at the subdistal appendages of the mother centrioles, this localization pattern suggests preferential trafficking of satellites along these microtubules [28]. Because satellite targeting to the cell periphery or center was dependent on microtubules, we next examined whether these satellite cluster(s), once they formed, were maintained independently of microtubules or not. The clusters remained mostly intact and did not split into smaller granules upon depolymerization of

microtubules by nocodazole treatment, suggesting that satellite granules fuse into stable solid-like clusters when forced into close proximity (Fig 1I, S3 Movie, S4 Movie). Together, these results show that rapamycin-induced dimerization of satellites with molecular motors irreversibly perturbs their distribution and size, and thus this assay provides a tool to investigate satellite functions in a temporally controlled manner.

## Satellites exert variable effects on pericentrosomal abundance of their residents

Previous work has shown that the cellular loss of satellites, by acute or chronic depletion of PCM1, alters the centrosomal levels and dynamics of a subset of proteins [5–8,29]. Given that satellites interact with and store a wide range of centrosome proteins, their pericentrosomal localization is likely required for efficient protein targeting to the centrosomes. To test this and to determine which proteins are regulated by satellites, we co-transfected HeLa cells with the indicated constructs and used quantitative immunofluorescence to measure the pericentrosomal levels of various known satellite proteins by measuring fluorescence signal intensities within a 3-$\mu$m$^2$ circle encompassing the centrosome. We chose satellite proteins that also localize to different parts of the centrosome (e.g., pericentriolar material, distal appendages, centriole linker) and are implicated in different centrosome/cilium-associated cellular functions (Fig 2A and 2B). As controls, we quantified the transfected cells in which GFP-PCM1-FKBP localized, like endogenous PCM1, and excluded the ones that were impaired for pericentrosomal satellite clustering due to overexpression of molecular motors. For quantifying centrosomal levels in cells with peripheral or centrosomal satellite clustering after rapamycin treatment, we only accounted for the cells that exhibited complete redistribution to the cell center or periphery.

The pericentrosomal levels of the ciliogenesis factors, including the transition zone component Cep290, the mindbomb E3 ubiquitin ligase 1 (Mib1), Cep131, and Cep72, increased significantly in centrosomally targeted BICD2-expressing cells and decreased significantly in peripherally targeted Kif5b-expressing cells at 6 and 24 hours after rapamycin treatment (Fig 2C, S2A Fig). We note that these proteins were also recruited to the PCM1-positive clusters at the cell periphery in Kif5b-expressing cells (Fig 2B and 2C). Pericentrosomal accumulation of other satellite proteins implicated in cilium assembly and centriole duplication, including KIAA0753, coiled-coil domain containing 14 (CCDC14), and oral-facial-digital syndrome 1 (OFD1), were affected similarly upon satellite mispositioning (S2B Fig). In contrast, there was no significant change in the pericentrosomal levels of the centriole duplication factors Cep63 and Cep152, the distal appendage protein Cep164, the centriole component centrin, and the centrosomal linker protein centrosomal Nek2-associated protein 1 (C-Nap1) in cells with peripheral satellite clustering (Fig 2B, S2A Fig). Additionally, except for centrin, these proteins were not recruited to the peripheral aggregates in Kif5b-expressing cells.

We next examined the potential role of satellite positioning in regulating the daughter centriole composition by using the rapamycin-treated BICD2-expressing cells, which leads to satellite accumulation around the mother centriole (Fig 1I). We chose the daughter centriole protein Cep120 for further study as it was identified in the satellite proteome, and 60% of its centrosomal pool was shown to be mobile [3,30,31]. We stained HeLa cells co-expressing GFP-PCM1-FKBP and HA-BICD2 before and after rapamycin addition with Cep120, the centrosome marker gamma-tubulin, and the mother centriole marker Cep164 (S3A Fig). Despite its enrichment in the daughter centriole in control cells, Cep120 was strongly enriched in the mother centriole in BICD2-expressing cells (S3A Fig). This indicates that satellite positioning regulates the enrichment of Cep120 at the daughter centriole.

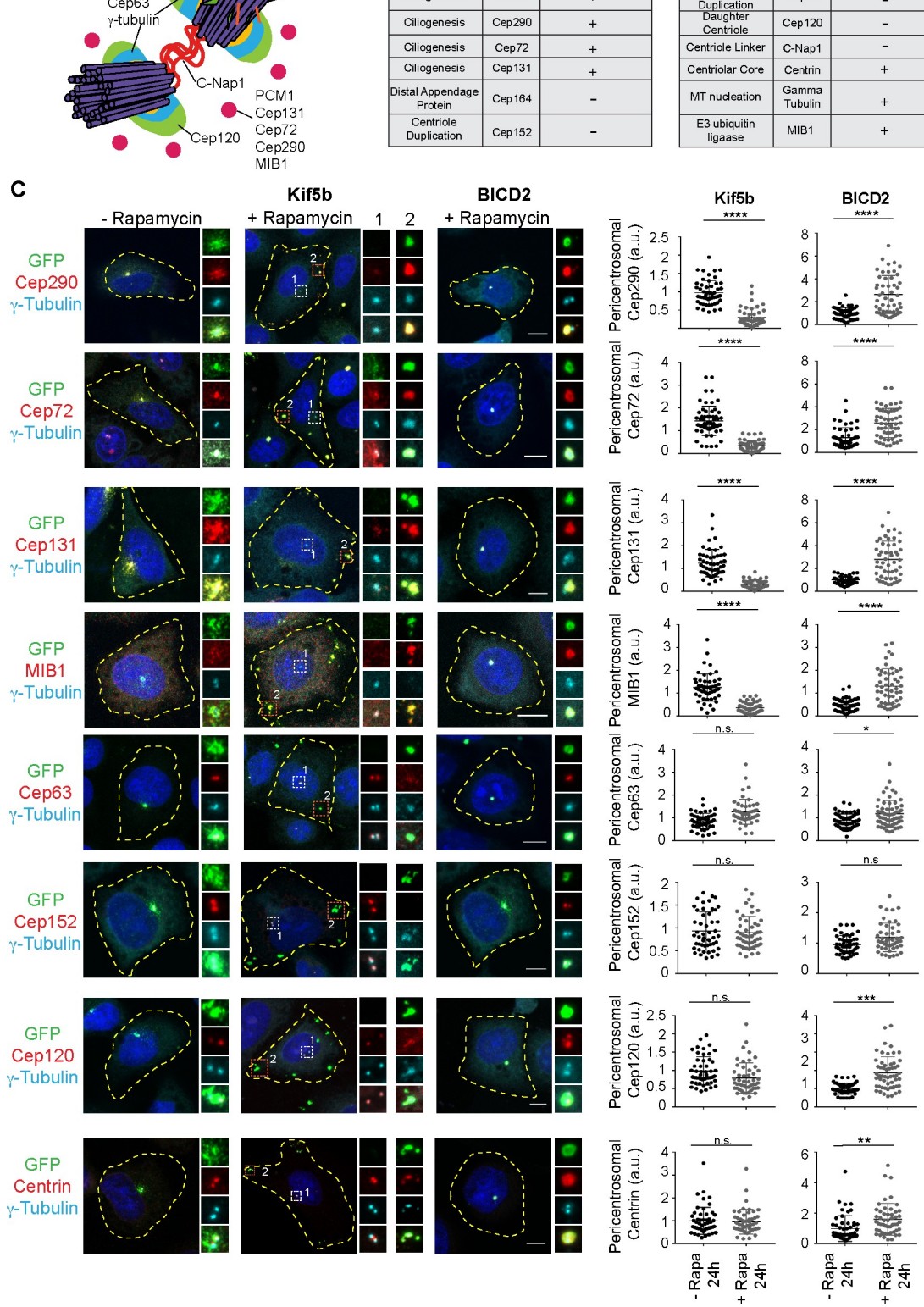

**Fig 2. Proper satellite distribution is required for pericentrosomal abundance of satellite residents at varying levels.** Effects of satellite redistribution to the cell periphery or center on pericentrosomal abundance of multiple satellite residents with varying functions and subcentrosomal localizations. **(A)** Spatial localization of the selected proteins at the centrosome. **(B)** Summary of the results for the changes in the pericentrosomal abundance of the indicated proteins and their associated functions and spatial localizations at the centrosome and cilia. "Localization at peripheral clusters" represents the group of proteins that concentrate with PCM1 at the periphery upon rapamycin addition to cells co-expressing GFP-PCM1-FKBP and HA-Kif5b-FRB. **(C)** HeLa cells co-expressing GFP-PCM1-FKBP with HA-Kif5b-FRB or HA-BICD2-FRB were treated with rapamycin for 1 hour followed by fixation at 24 hours. Cells that were not treated with rapamycin were processed in parallel as controls. Cells were stained with anti-GFP to identify cells with complete redistribution to the cell periphery or center, and anti-gamma-tubulin to mark the centrosome and antibodies against the indicated proteins. Images represent centrosomes in cells from the same coverslip taken with the same camera settings. DNA was stained by DAPI. Fluorescence intensity at the centrosome was quantified, and average means of the levels in control cells at 6 hours were normalized to 1. $n \geq 25$ cells per experiment. Data represent mean value from two experiments per condition, ±SD (***$p < 0.001$, ****$p < 0.0001$). Cell edges are outlined. Scale bars, 10 μm, all insets show 4× enlarged centrosomes. Error bars = SD. Source data can be found in S2 Data. a.u., arbitrary unit; BICD2, bicaudal D homolog 2; C-Nap1, centrosomal Nek2-associated protein 1; FKBP, FK506 binding protein 12; FRB, FKBP12-rapamycin-binding; GFP, green fluorescent protein; HA, hemagglutinin; Kif5b, kinesin family member 5b; MIB1, mindbomb E3 ubiquitin ligase 1; n.s., nonsignificant; PCM1, pericentriolar material 1; Rapa, rapamycin.

Satellite interactome is highly enriched in microtubule-associated proteins, including the key regulators of microtubule nucleation and dynamics, such as gamma-tubulin and the components of the HAUS/Augmin complex [2,3]. While the centrosomal gamma-tubulin levels did not change in cells with peripheral satellite clustering, gamma-tubulin concentrated at the peripheral satellite clusters (S3B Fig). To test whether recruitment of gamma-tubulin to these clusters results in microtubule nucleation, we performed microtubule regrowth experiments in inner medullary collecting duct (IMCD3) cells stably expressing GFP-PCM1-FKBP and HA-Kif5b. After inducing peripheral accumulation of satellites by rapamycin, microtubules were depolymerized by nocodazole treatment. Following nocodazole washout, cells were fixed and stained for alpha-tubulin at the indicated time points (S3C Fig, S3D Fig). Despite prominent microtubule nucleation at the centrosomes, microtubule nucleation was not initiated at the peripheral gamma-tubulin–positive clusters. Thus, the concentration of gamma-tubulin at the peripheral satellites does not assign them microtubule organizing center (MTOC) activity, which might be due to lack of modifications or regulatory proteins required for microtubule nucleation.

## The majority of the satellite interactome remained unaltered upon peripheral targeting

To determine whether and, if so, how the composition of satellites changed upon their peripheral targeting, we took a systematic approach. To this end, we applied the Biotin Identification (BioID) proximity labeling approach and identified the PCM1 proximity interactome before and after rapamycin treatment in Kif5b-expressing cells. PCM1-FKBP was fused to Myc-BirA* at the N terminus and co-expressed with HA-Kif5b in HEK293T cells. Myc-BirA*-PCM1-FKBP localized to and induced biotinylation at the satellites, as assessed by staining for Myc, streptavidin, and gamma-tubulin (Fig 3A). As expected, localized biotinylation was observed at the pericentrosomal granules in control cells and at the peripheral satellite clusters in rapamycin-treated cells (Fig 3A). Given that satellites are biotinylated at the periphery away from the centrosomes, their proteome will likely exclude the contaminating interactions of PCM1 with the centrosome due to their close proximity.

For mass spectrometry experiments, HEK293T cells were co-transfected with HA-Kif5b and Myc-BirA*-PCM1-FKBP, treated with rapamycin 24 hours post-transfection for 1 hour, and incubated with biotin for 18 hours. Cells that were not treated with rapamycin and cells expressing Myc-BirA* only were used as controls. Efficient streptavidin pulldown of Myc-BirA*-PCM1-FKBP and biotinylated proteins were confirmed by western blotting (Fig 3B).

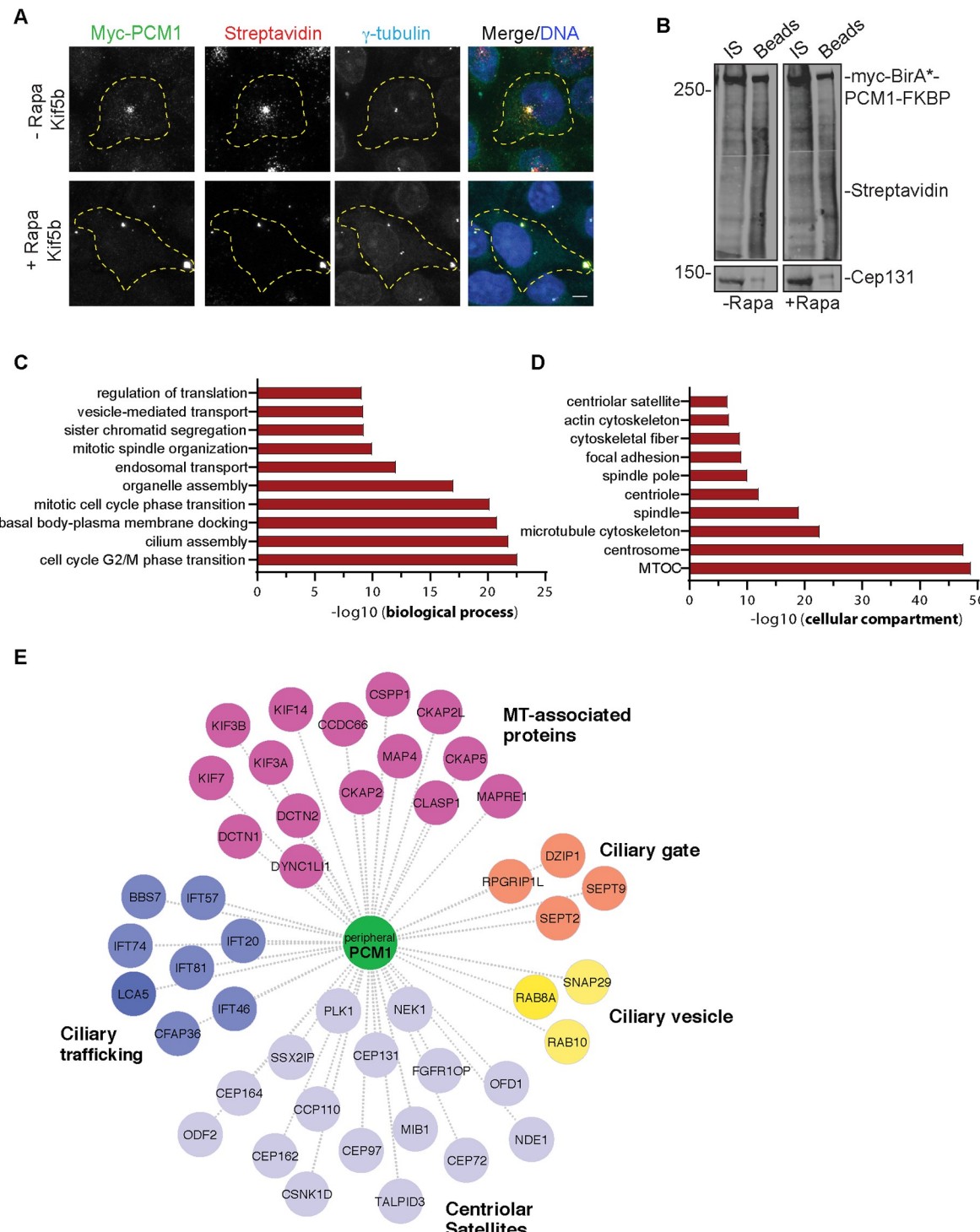

**Fig 3. The proximity interactome of PCM1 at the peripheral clusters is enriched for proteins implicated in ciliogenesis and mitosis.** The proximity PCM1 interactome of satellites were identified using the BioID approach. (**A**) HEK293T cells were transfected with Myc-BirA*-PCM1-FKBP and induced for peripheral targeting of satellites with rapamycin treatment for 1 hour. Cells that were not treated with rapamycin were used as a control. After 18 hours biotin incubation, cells were fixed and stained for Myc-BirA*-PCM1-FKBP expression with anti-Myc, biotinylated proteins with streptavidin, and centrosomes with gamma-tubulin. DNA was stained with DAPI. Scale bar, 10 μm; cell edges are outlined. (**B**) Biotinylated proteins from lysates from cells expressing Myc-BirA*-PCM1-FKBP (−rapamycin or +rapamycin) were pulled down with streptavidin chromatography and samples were analyzed by SDS-PAGE and western blotting with streptavidin to detect biotinylated proteins and with anti-Cep131 (positive control). IS, initial sample used for streptavidin pulldowns;

Beads, biotinylated and captured proteins. **(C, D)** GO-enrichment analysis of the proximity interactors of PCM1 after rapamycin treatment based on their **(C)** biological processes and **(D)** cellular compartment. The x-axis represents the log-transformed *p*-value (Fisher's exact test) of GO terms. Source data can be found in S4 Data. **(E)** The cilium-associated proteins in the interactome of peripheral satellites were determined based on GO-enrichment analysis and previous studies. Different functional modules of ciliogenesis were plotted in the "peripheral PCM1 interaction network" using Cytoscape. BioID, Biotin Identification; FKBP, FK506 binding protein 12; GO, Gene Ontology; MT, microtubule; PCM1, pericentriolar material 1; Rapa, rapamycin.

Biotinylated proteins from two technical and two experimental replicates for each condition were identified by mass spectrometry and analyzed by significance analysis of interactome (SAINT) to identify high-confidence interactions (S1 Table). SAINT analysis identified 541 proximity interactions for PCM1 before rapamycin treatment and 601 interactions after rapamycin treatment (Bayesian false discovery rate [BFDR] <0.01) (S2 Table). The control and peripheral PCM1 interactomes shared 476 components, while 65 proteins were specific to peripheral satellites and 125 proteins were specific to control cells (S4A Fig). The proteins enriched (>1.25-fold relative to control) or depleted (<0.75-fold relative to control) at the peripheral satellites, with their associated Gene Ontology (GO) biological processes, are shown in S4 Fig. The 71% overlap before and after rapamycin treatment shows that the composition of the PCM1 proximity interactome remains mostly unaltered upon peripheral targeting of satellites. The peripheral PCM1 interactome is significantly enriched in previously identified MTOC, centrosome, satellite, and microtubule cytoskeleton components based on GO analysis (Fig 3C and 3D). Of note, we did not identify enrichment for components of microtubule cargoes such as endosomes and lysosomes, which shows that the effects of the mispositioning assay are specific to satellites (Fig 3D).

GO analysis of the proteins identified in the PCM1 interactome of peripheral satellites for biological processes revealed significant enrichment for processes related to cilium assembly and mitotic progression (Fig 3C, S3 Table). Given that satellite-less epithelial cells were mainly defective in cilium assembly, we aimed to use this dataset to gain insight into the specific ciliary processes regulated by satellites [7,8]. To this end, we determined the PCM1 proximity interactors that were previously implicated in primary cilium biogenesis and functions [32,33], and generated an interaction network by classifying them based on their ciliary functions and interactions (Fig 3E). Among the highly represented categories are the microtubule-associated proteins previously localized to cilia such as coiled-coil domain containing 66 (CCDC66), centrosome and spindle pole associated protein 1 (CSPP1), and microtubule-associated protein 4 (MAP4) and centrosome/satellite proteins that regulate cilium biogenesis and function. Additionally, ciliary trafficking complexes IFT-B, kinesin 2, and dynein; proteins implicated in ciliary vesicle formation, septins, and transition zone components were also identified in the proteome of peripheral satellite clusters. Collectively, systematic analysis of the composition of peripheral satellites suggest cilium- and mitosis-related cellular processes as potential satellite functions.

## Peripheral centriolar satellite clustering results in defective cilium assembly, maintenance, and disassembly

Given that our results thus far showed that satellites and associated centrosome proteins can be targeted to the periphery rapidly and efficiently in an inducible way, we employed this assay to investigate temporal satellite functions in ciliated and proliferating cells. Satellite-less PCM1$^{-/-}$ kidney and retina epithelial cells had significant defects in their ability to assemble primary cilia [7,8]. We therefore first examined whether pericentrosomal clustering of satellites is also required for cilium assembly. To this end, we generated IMCD3 and IMCD3::PCM1$^{-/-}$ cells that stably express GFP-PCM1-FKBP and HA-Kif5b-FRB (hereafter

IMCD3[peripheral] and IMCD3 PCM1 knockout [KO][peripheral] cells). Rapamycin induction resulted in localization of satellites at the cell periphery in both cell lines (Fig 4A, S5A Fig). Expression of GFP-PCM1-FKBP restored the ciliogenesis defect of IMCD3 PCM1 KO cells, confirming that this fusion protein is fully functional (IMCD3 PCM1 KO: 56% ± 2.9%, IMCD3 PCM1 KO[peripheral]: 67.6% ± 2.1%) (S5B Fig). We were unable to generate cells stably co-expressing GFP-PCM1-FKBP and HA-BICD2-FRB, likely due to the lethality of constitutive BICD2 expression and subsequent inhibition of dynein activity. Therefore, we investigated the consequences of perturbing satellite proximity to the centrosome on cilium assembly only in cells with peripheral satellite clustering. To this end, IMCD3[peripheral] and IMCD3 PCM1 KO[peripheral] cells were treated with rapamycin for 1 hour, serum-starved for 48 hours, and the percentage of ciliated cells was quantified by staining cells for ADP ribosylation factor-like GTPases 13b (Arl13b), a marker for the ciliary membrane, and acetylated tubulin, a marker for the ciliary axoneme (Fig 4B). While control cells ciliated at 67.6% ± 1.1%, IMCD3 PCM1 KO[peripheral] cells ciliated at 48.5% ± 5% ($p < 0.01$) (S5B Fig). Notably, the cilia that formed in these cells were also significantly shorter in length (−Rapa: 2.9 μm ± 1, +Rapa: 2.5 μm ± 0.92, $p < 0.01$) (Fig 4C). Similarly, IMCD3[peripheral] cells had a significant reduction in their ciliation efficiency relative to control cells (control: 77.35% ± 2.3%, IMCD3[peripheral]: 54.4% ± 2.0%) ($p < 0.01$) (S5C Fig). As a control for the possible effects of rapamycin on cilium assembly, we performed these experiments in IMCD3 cells treated with rapamycin and found that the concentrations and incubation times we used did not affect ciliogenesis efficiency ($p = 0.87$) (S5D Fig).

To gain insight into the mechanisms underlying the ciliogenesis defects of IMCD3 PCM1 KO[peripheral] cells, we examined their cilium assembly dynamics by quantifying percentage of ciliated cells at different time points post serum starvation. Peripheral satellite clustering resulted in a significant decrease in the fraction of ciliated cells only 16, 24, and 24 hours after serum starvation (Fig 4D). The lack of ciliogenesis defects in rapamycin-treated cells at 2 and 6 hours post serum starvation suggests that proper satellite positioning is not required for the early stages of ciliogenesis in IMCD3 cells, such as recruitment of periciliary/ciliary vesicles to the mother centriole and mother centriole maturation [34].

The inducible nature of the satellite trafficking assay enabled us to study the temporal function of satellites in ciliated cells during primary cilium maintenance and disassembly, which could not have been tested in satellite-less PCM1[−/−] cells [7,8]. Given that IMCD3[peripheral] and IMCD3 KO[peripheral] cells behaved similarly in ciliogenesis experiments, we performed these experiments only in IMCD3[peripheral] cells. To study the function of satellites in regulation of steady-state cilia stability, cells were serum starved for 48 hours, treated with rapamycin for 1 hour, and the percentage of cilia was quantified in a time-course manner after rapamycin washout (Fig 4E). As expected, the percentage of ciliation did not change over 24 hours in control cells (Fig 4E, S5E Fig). However, induction of peripheral satellite clustering resulted in a 22% decrease in the percentage of ciliated cells at 6 hours after rapamycin treatment (control: 69% ± 0.5%, Rapa: 47.6% ± 3.3%, $p < 0.0001$). The fraction of ciliated cells continued to decrease over time until only 38% ± 1% of cells retained primary cilia after 24 hours of rapamycin treatment, compared to the stable 69.5% ± 0.5% of control cells ($p < 0.0001$) (Fig 4E, S5E Fig). Rapamycin treatment had no effect on maintenance of control cells (S5D Fig). Next, we examined whether satellite mispositioning can affect cilium disassembly after serum restimulation. To this end, cells were serum starved for 48 hours, treated with rapamycin for 1 hour, and the percentage of cilia was quantified over a 6-hour serum-stimulation time course (Fig 4F, S5F Fig). In control cells, the percentage of ciliation decreased from 68.5% ± 0.5% to 56% ± 1% at 2 hours, 46% ± 0% at 4 hours, and to 38% ± 4.5% at 6 hours (Fig 4F, S5F Fig). In rapamycin-treated cells, there was a rapid and significantly greater decrease in the fraction of ciliated

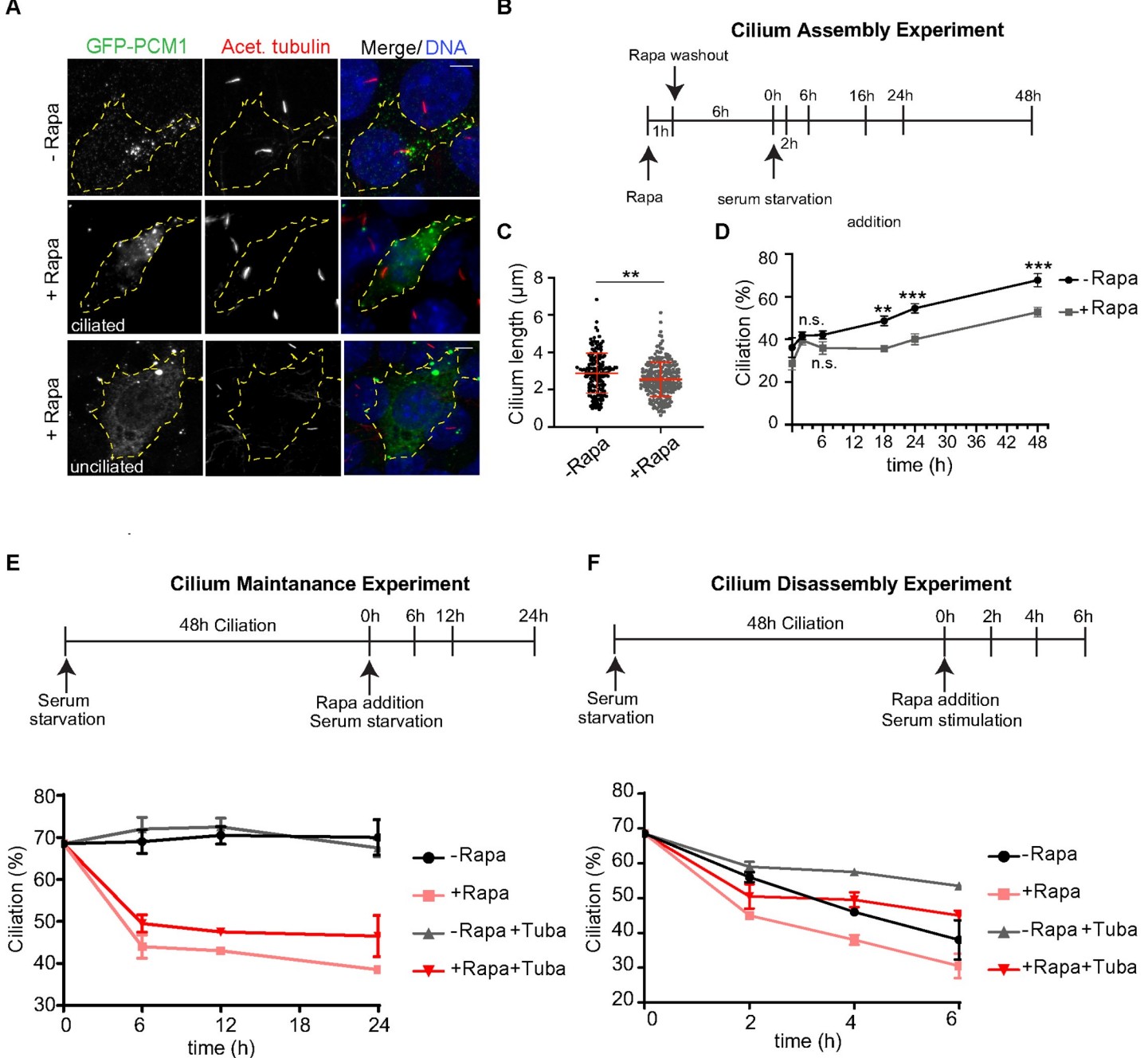

**Fig 4. Pericentrosomal satellite clustering is required for efficient cilium assembly, maintenance, and disassembly. (A, B, C)** Effect of peripheral satellite clustering on cilium assembly dynamics and cilium length. **(A)** Representative images of ciliated and unciliated cells with peripheral satellite clustering relative to ciliated control IMCD3 PCM1 KO[peripheral] cells. Cell edges are outlined. Scale bar, 10 μm. **(B)** The experimental protocol for assaying cilium assembly dynamics by immunofluorescence analysis. Control and rapamycin-treated IMCD3 PCM1 KO[peripheral] cells stably expressing GFP-PCM1-FKBP and HA-Kif5b-FRB were induced for satellite redistribution to periphery by 1 hour after rapamycin treatment. After rapamycin washout, cells were incubated in complete medium for 5 hours and serum-starved for the indicated time points. The percentage of ciliated cells was determined by staining for acetylated tubulin in cells with complete satellite redistribution to the periphery as assessed by GFP staining. **(C)** Quantification of cilium length in control and rapamycin-treated IMCD3 PCM1 KO[peripheral] 48 hours post serum starvation. Results shown are the mean of two independent experiments ± SD (>50 cells/experiment, $^{**}p < 0.01$). **(D)** Quantification of percentage of ciliated control and rapamycin-treated IMCD3 PCM1 KO[peripheral] cells before serum starvation and at 2, 6, 16, 24, and 48 hours after serum starvation. Results shown are the mean of two independent experiments ± SD (>50 cells/experiment, $^{**}p < 0.01$, $^{***}p < 0.001$, n.s., nonsignificant). **(D)** Effect of peripheral satellite clustering on cilium maintenance. Cells were serum-starved for 48 hours, treated with rapamycin for 1 hour, and percentage of ciliated cells was determined over 24 hours by staining for acetylated tubulin. Cells that were not treated with rapamycin were used as a control. The same experiments were also performed in the presence of 2 μM tubacin. Data represent the mean value from two experiments per condition. **(F)** Effect of peripheral satellite clustering on cilium disassembly. Cells were serum-starved for 48 hours,

treated with rapamycin for 1 hour, induced by serum-stimulation, and the percentage of ciliated cells was determined over 6 hours by staining for acetylated tubulin. Cells that were not treated with rapamycin were used as a control. The same experiments were also performed in the presence of 2 μM tubacin. Data represent mean value from two experiments per condition. Error bars = SD; source data can be found in S5 Data. Acet., acetylated tubulin; FKBP, FK506 binding protein 12; FRB, FKBP12-rapamycin-binding; GFP, green fluorescent protein; HA, hemagglutinin; IMCD3, inner medullary collecting duct; Kif5b, Kinesin family member 5b; KO, knockout; PCM1, pericentriolar material 1; Rapa, rapamyccin; Tuba, tubacin.

cells at 2 hours, from 68.5% ± 0.5% to 45% ± 0% (Fig 4F, S5F Fig), which corresponds to the first wave of cilium disassembly [35,36]. After 6 hours post serum addition, only 30.5% ± 2.5% of cells with peripheral satellite localization had cilia, compared to 38% ± 4% of control cells ($p = 0.15$) (Fig 4F, S5F Fig). The enhanced deciliation and disassembly phenotypes upon peripheral satellite clustering identified satellites as regulators of cilium maintenance and disassembly in addition to their reported functions in assembly.

The maintenance and disassembly of cilium requires modifications of the axonemal tubulins [34]. A major event that promotes cilium disassembly is the phosphorylation of histone deacetylase 6 (HDAC6) by Aurora A kinase and subsequent deacetylation of the modified tubulins of the ciliary axoneme and cortactin [37,38]. To test whether satellites regulate cilium maintenance and disassembly in a HDAC6-dependent manner, we performed cilium maintenance and disassembly experiments in control and rapamycin-treated cells using tubacin as a potent and selective inhibitor of HDAC6 deacetylase activity [38]. Analogous to control cells, tubacin treatment rescued the enhanced cilium disassembly phenotype of rapamycin-treated cells at 4 and 6 hours (4 hours: Rapa: 38% ± 4.5%, tubacin+Rapa: 49.5% ± 1.5% ($p < 0.01$); 6 hours: Rapa: 30.5% ± 2.5%, tubacin+Rapa: 45% ± 1%, $p < 0.01$), but did not rescue the cilium maintenance defects at 6,12, or 24 hours ($p > 0.05$) (Fig 4E and 4F). These results show differential requirements for the regulation of cilium maintenance and disassembly, and suggest that satellites function in cilium disassembly by regulating the structural integrity of the ciliary axoneme upstream of HDAC6.

## Centrosomal and peripheral satellite clusters do not dissolve during mitosis, and display differential mitotic phenotypes

Satellite integrity and localization is dynamically modulated during cell division, and the functional significance of these changes remains poorly understood [6,13]. The enrichment of key regulators of microtubule nucleation and dynamics and mitosis such as nuclear mitotic apparatus (NUMA), polo-like kinase 1 (Plk1), and cyclin-dependent kinase 1 (CDK1) in the satellite proteome suggests functions for satellites during mitosis [2,3]. Unexpectedly, satellite-less IMCD3 and retinal pigmented epithelial (RPE1) PCM1 KO cells did not exhibit defects in cell proliferation, cell cycle progression, and mitotic times, which could be due to the activation of compensatory mechanisms [7]. To address this possibility and to investigate whether and, if so, how satellites contribute to mitosis, we assayed the mitotic behavior of cells upon rapamycin induction of satellite redistribution. First, we examined the mitotic dynamics of centrosomal and peripheral satellite clusters using immunofluorescence and time-lapse imaging. In control HeLa cells co-expressing GFP-PCM1-FKBP with HA-Kif5b, as expected, satellites dissolved during mitosis, which was accompanied by an increase in the soluble cytoplasmic pool of PCM1 (Fig 5A). In rapamycin-treated cells, the satellite clusters at the periphery did not dissolve during mitosis and localized to the cell periphery during metaphase and anaphase, followed by accumulation at the cytokinetic bridge (Fig 5A). Similar mitotic behavior was observed in IMCD3 PCM1 KO[peripheral] cells after rapamycin treatment, as assessed by time-lapse imaging (Fig 5B). In rapamycin-treated HeLa cells co-expressing HA-BICD2 and GFP-PCM1-FKBP, the satellite clusters at the centrosome also did not dissolve and remained

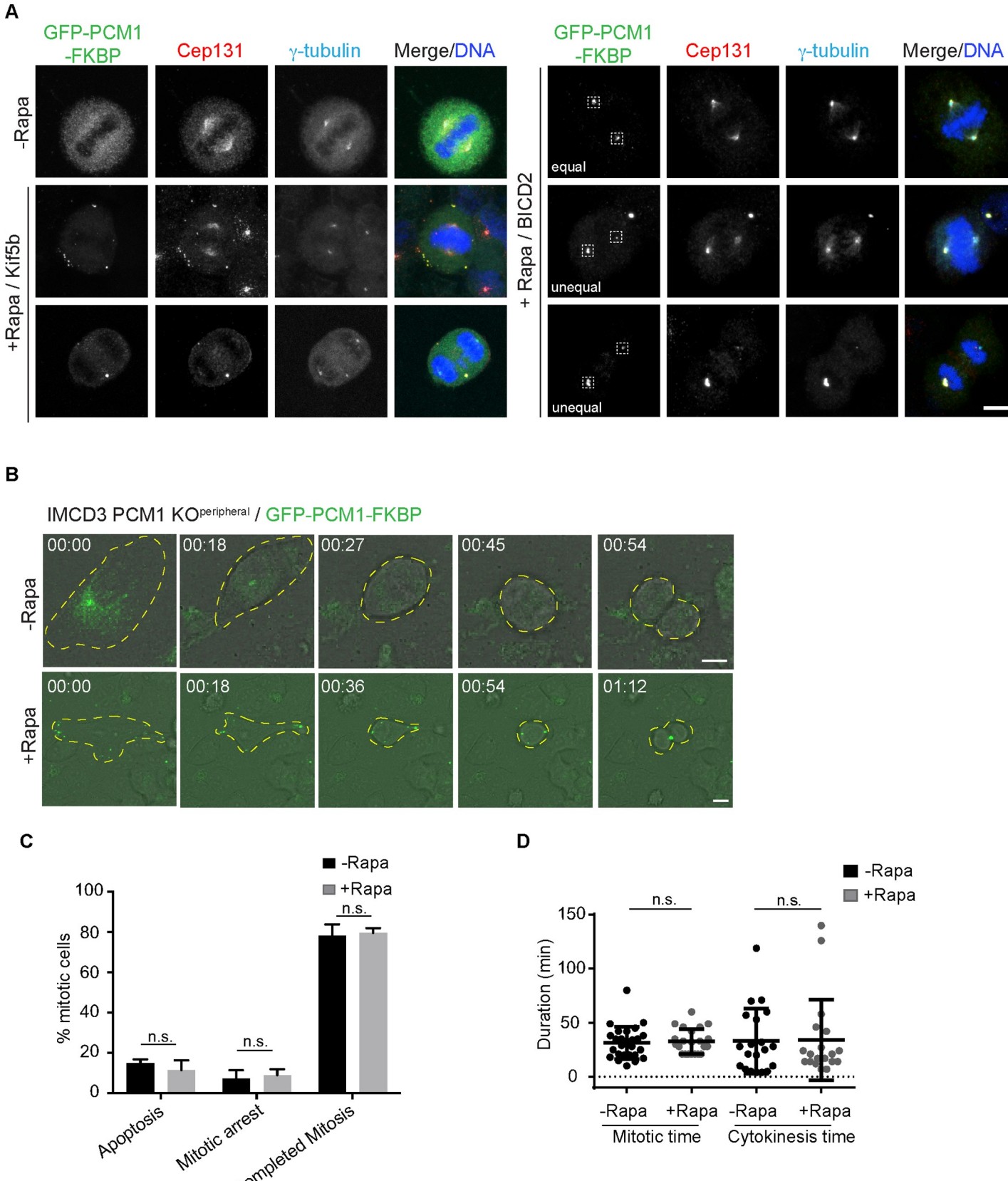

**Fig 5. Peripheral satellite clusters persist during mitosis and do not compromise mitotic progression.** **(A)** Effect of satellite clustering at the cell periphery and the center on satellite integrity and dynamics during mitosis. Control and rapamycin-treated HeLa cells were fixed 24 hours after rapamycin treatment and stained for GFP, Cep131, gamma-tubulin, and DAPI. Dashed boxes indicate the spindle poles; "unequal" indicates unequal segregation of satellite clusters between spindle poles and "equal" indicates equal segregation. Scale bar, 10 μm. **(B)** Effect of peripheral satellite clustering on mitotic dynamics of satellites and mitotic progression. IMCD3 PCM1 KO$^{peripheral}$ cells were imaged every 4.5 minutes for 16 hours. Satellite clusters did not dissolve during mitosis in rapamycin-treated cells but dissolved in control cells. Representative brightfield and fluorescence still frames from time-lapse experiments were shown. Cell edges are outlined. Scale bar, 10 μm. **(C)** Effect of peripheral satellite clustering on mitotic progression. Percentage of control and rapamycin-treated mitotic cells were imaged by the time-lapse imaging, and videos were analyzed to classify mitotic cells into categories of cells that completed mitosis, cells that underwent apoptosis, and cells arrested in mitosis for more than 5 hours. $n > 20$ cells per experiment. Data represent mean ± SD of three independent experiments. **(D)** Effect of peripheral satellite clustering on duration of mitosis and cytokinesis. Mitotic time was quantified as the time interval from nuclear envelope breakdown to anaphase onset. Cytokinesis time was quantified as the time from anaphase onset to cytokinetic cleavage. $n > 20$ mitotic cells per experiment. Data represent mean ± SD of two independent experiments. Error bars = SD. Source data can be found in S6 Data. FKBP, FK506 binding protein 12; GFP, green fluorescent protein; IMCD3, inner medullary collecting duct; Kif5b, kinesin family member 5b; KO$^{peripheral}$, knockout$^{peripheral}$; ns, nonsignificant; PCM1, pericentriolar material 1; Rapa, rapamycin.

associated with the spindle poles (Fig 5A). Notably, a fraction of cells had satellite clusters at both poles, while the rest had the cluster only in one pole (Fig 5A). These results together show that the satellite clusters at the centrosome and periphery do not undergo mitotic dissolution.

To examine the phenotypic consequences of satellite mispositioning and inhibition of their mitotic dissolution, we analyzed the movies before and after rapamycin treatment to quantify the percentage of mitotic cells that (1) completed mitosis, (2) underwent apoptosis, (3) arrested in mitosis for more than 5 hours. In IMCD3 PCM1 KO$^{peripheral}$ cells with peripheral satellite clustering, there was not significant change in the fraction of mitotic cells that arrested in mitosis or underwent apoptosis, as assessed by membrane blebbing and DNA fragmentation (3 independent experiments, $p > 0.05$) (Fig 5C). Next, we quantified the mitotic and cytokinesis time for the group of control and rapamycin-treated IMCD3 PCM1 KO$^{peripheral}$ cells that completed mitosis during the time of imaging. Mitotic time was determined as the time from nuclear envelope breakdown to anaphase onset by using mCherry-H2B dynamics or brightfield images as references. While the average mitotic time was 31.4 ± 2.9 minutes for IMCD3 PCM1 KO$^{peripheral}$ cells that are not treated with rapamycin ($n = 26$), it was 32.8 ± 2.4 for IMCD3 PCM1 KO$^{peripheral}$ cells after rapamycin treatment ($n = 27$) (2 independent experiments, $p > 0.05$) (Fig 5D). Cytokinesis time was quantified as the time frame between anaphase onset and cleavage, and it was similar between control and rapamycin-treated cells (−Rapa: 32.9 ± 6.7 minutes [$n = 20$], +Rapa: 48.6 ± 12.9 [$n = 27$]) (Fig 5D). Rapamycin treatment by itself did not affect mitotic and cytokinesis times (S6A Fig).

In parallel to characterization of IMCD3 PCM1 KO$^{peripheral}$ stable cells, we investigated the mitotic phenotypes associated with peripheral or centrosomal satellite clustering in HeLa cells co-expressing GFP-PCM1-FKBP with HA-Kif5b or HA-BICD2. Analogous to rapamycin-treated IMCD3 PCM1 KO$^{peripheral}$ cells, HeLa cells with peripheral satellite clusters did not have defects in mitotic progression (S6B and S6C Fig) and had similar mitotic times as compared with control cells (S6E and S6F Fig). In contrast, HeLa cells with centrosomal satellite clusters had significant defects both in mitotic progression and mitotic times. Specifically, we found a significant decrease in the fraction of cells that completed mitosis (−Rapa: 85.5% ± 4.8%, +Rapa: 14.8% ± 10.8%, $p < 0.01$) (S6B and S6D Fig). After a prolonged mitotic arrest, 63.3% ± 9.8% of these cells underwent apoptosis relative to the 8.4% ± 5.7% control apoptotic cells ($p < 0.01$) (S6B and S6D Fig). Finally, the average mitotic times were 37.8 ± 1.7 minutes for control cells ($n = 36$ mitotic cells) and 114.3 ± 15.3 for BICD2-expressing cells ($n = 26$ cells) ($p < 0.0001$), as assessed by dynamics of chromosomes stained with silicone rhodamine (SIR)-Hoechst dye (S6E and S6F Fig). Collectively, these results show that proper satellite distribution and remodeling is required for mitotic progression.

## Discussion

We developed a chemically inducible satellite trafficking assay that allowed us to rapidly and specifically target satellites to the cell periphery and used this new tool to study satellite functions in a temporally controlled way in ciliated and proliferating cells. Application of this assay in epithelial cells for the first time, to our knowledge, identified functions for satellites not just for cilium assembly but also for cilium maintenance, cilium disassembly, and mitosis. Targeted and systematic quantification of the satellite proteome at the cell periphery suggested defective centrosomal targeting of proteins implicated in these processes as possible mechanisms underlying these phenotypes. In addition to probing temporal satellite functions, our results for the first time, to our knowledge, also revealed a direct link between satellite functions and their cellular positioning. Importantly, we note that the satellite-trafficking assay provides a powerful tool for future studies aimed at investigating cell type–specific functions and mechanisms of satellites. Its use in specialized cell types, in particular the ones with non-centrosomal MTOCs such as differentiated muscle cells and polarized epithelial cells, has the potential to provide new insight into why and how satellite distribution varies in different contexts.

Inducible peripheral satellite targeting in ciliated cells revealed novel functions for satellites as positive regulators of cilium maintenance and disassembly. Cilium disassembly upon serum stimulation in IMCD3 cells occurs predominantly by whole-cilium shedding, which requires HDAC6 activity [35]. Enhanced cilium disassembly defect of rapamycin-induced IMCD3[peripheral] cells was partially restored by HDAC6 inhibition, suggesting that satellites function upstream of HDAC6. Because HDAC6 was not identified in the satellite interactome, it is likely that satellites are not direct regulators of HDAC6 activity and that they might take part in cilium disassembly through regulating other disassembly regulators such as PLK1 [2,3]. Of note, PLK1 is part of the peripheral satellite proteome we generated in this study. Importantly, tubacin treatment did not rescue the cilium maintenance defect, highlighting differential regulation of cilium maintenance and disassembly through independent and/or overlapping mechanisms. Given that we identified the IFT-B components including IFT20, IFT81, IFT57, IFT46, and IFT74 in the proteome of peripheral satellites in Kif5b-expressing cells, we propose that perturbation of IFT trafficking through sequestration of IFT components at the peripheral satellite clusters might underlie the cilium maintenance defects. Supporting this mechanism, sequestration of IFT20 and IFT74 at the mitochondria using the rapamycin-dimerization approach in ciliated fibroblasts resulted in a similar defect in cilium maintenance [19]. Defective IFT-B complex targeting could also explain the cilium assembly and shorter cilia phenotypes associated with peripheral satellite clustering. Together with previous work that identified regulatory roles for satellites in IFT88 targeting to the basal body and cilia [7], our results indicate the presence of a critical but poorly understood regulatory relationship between satellites and the IFT-B complex. Whether IFT-B complex is regulated at the assembly, activity and/or targeting level by satellites remain as critical outstanding questions.

Satellites are remodeled under certain physiological conditions, and the cell cycle is the best-characterized context for studying satellite remodeling [39]. A recent study identified DYRK3 as the key regulator of mitotic dissolution of multiple membraneless organelles, including satellites, and showed that its inhibition resulted in prolonged mitosis, mitotic arrest, and multipolar spindle formation [13]. Our results revealed that the satellite clusters formed in the cell periphery or the center in rapamycin-treated cells did not dissolve during mitosis, which might be due to fusion of satellites into solid-like aggregates or to inhibition of DYRK3 activity. Notably, cells with centrosomal satellite clusters, but not with peripheral clusters, displayed mitotic phenotypes. The mitotic phenotypes cannot be explained by inhibition of the

mitotic dissolution of satellite clusters, and future studies are required to determine how satellite clustering at the centrosome compromises mitotic progression.

Depletion or deletion of various centrosome proteins perturbs cellular positioning of satellites either by inducing their dispersal throughout the cytoplasm or tighter concentration at the centrosome [5,39]. The satellite distribution defects have been proposed to underlie the phenotypes associated with loss of these proteins, such as defective ciliogenesis. However, these loss-of-function studies remained insufficient in directly testing this proposed link. The results of our study provide direct evidence for the functional significance of proper satellite positioning during primary cilium-associated processes and thus strengthen the conclusions of the previous work on how perturbed satellite localization might contribute to the function of their residents.

Satellites are proposed to mediate their functions in the context of centrosomes and cilia by trafficking proteins to or away from the centrosome and by sequestering centrosome proteins to limit their recruitment at the centrosome [5,39]. Using quantitative analysis of pericentrosomal levels of multiple centrosome proteins upon satellite redistribution and proximity mapping of peripheral satellite clusters, we found that only a subset of proteins required proper satellite distribution for their centrosomal localization. While these results provide further support to the functions of satellites in protein targeting, they also raise questions that pertain to why and how satellites do not regulate all their residents at this level. To gain insight into the rules that govern satellite selectivity, we categorized proteins regulated by satellites based on their evolutionary conservation pattern, functions, subcentrosomal localization, and interactions. These analyses did not reveal a tight relationship between these properties and satellite-mediated regulation. The only relationship that stood out was the one between proteins that interact and regulate the same cellular process, such as the CCDC14-KIAA0753 centriole duplication module and Cep72-Cep290 ciliogenesis module [40–42]. Co-regulation of these proteins suggests functions for satellites in mediating assembly of these complexes and/or their targeting to the centrosome as a complex.

Microtubules and molecular motors were shown to cooperate with satellites to regulate active protein targeting of satellite residents to the centrosome [5]. Because our results revealed a new functional relationship between proper pericentrosomal satellite positioning and centrosome/cilia-mediated processes, we propose a complementary way by which microtubules and motors regulate satellite-mediated functions. Our results suggest that they cooperate to maintain satellite proximity around the centrosome and that this is required for timely and efficient exchange of proteins between centrosomes and satellites. Notably, satellite granules fuse and form stable aggregates when forced into close proximity at the cell center or periphery [15], suggesting that cells have mechanisms to inhibit aggregation of satellites. Future studies are required to address several key questions that pertain to satellite mechanisms and cellular distribution: What is the precise mechanism that establishes and maintains pericentrosomal satellite clustering and inhibit their aggregation? Do satellites exchange material with the centrosome through active transport or simple diffusion? What are the upstream signaling pathways that induce protein release from satellites?

The direct link between cellular satellite positioning and their functions revealed by this study corroborates the notion that functions associated with different cellular compartments are regulated by their distinct spatial distribution within cells [43]. This relationship has been reported for multiple organelles and functions. One example is the nutrient-induced peripheral and starvation-induced perinuclear positioning of lysosomes, which was proposed to coordinate mTOR activity and autophagosome biogenesis during cellular anabolic and catabolic responses [44]. Another example is the requirement for proper mitochondrial positioning during T-cell activation to allow sustained local $Ca^{+2}$ influx and during neuronal differentiation to allow terminal axon branching [45–47].

## Materials and methods

### Cell culture, transfection, and transduction

Human cervical cancer HeLa and human embryonic kidney HEK293T cells were cultured with Dulbecco's Modified Eagle Medium (DMEM) medium (Pan Biotech, Cat. # P04-03590, Germany) supplemented with 10% fetal bovine serum (FBS) and 1% penicillin-streptomycin. Mouse kidney medulla collecting duct cells IMCD3:Flip-In cells were cultured with DMEM/F12 50/50 medium (Pan Biotech, Cat. # P04-41250, Germany) supplemented with 10% FBS (Life Technologies, Ref. #10270–106, Lot #42Q5283K, Carslbad, CA) and 1% penicillin-streptomycin (Gibco, Cat. # 1540–122, Gaithersburg, MD). All cell lines were authenticated by Multiplex Cell Line Authentication (MCA) and were tested for mycoplasma by MycoAlert Mycoplasma Detection Kit (Lonza, Germany).

IMCD3 and HeLa cells were transfected with the plasmids using Lipofectamine 2000 according to the manufacturer's instructions (Thermo Fisher Scientific, Waltham, MA). For serum starvation experiments, IMCD3 cells were washed twice with PBS and incubated with DMEM/F12 supplemented with 0.5% FBS for the indicated times. For cilium maintenance experiments, IMCD3 cells were serum starved for 48 hours, incubated with rapamycin for 1 hour, and fixed at the indicated times. For serum stimulation experiments, ciliated IMCD3 cells were incubated with rapamycin for 1 hour, then with DMEM/F12 50/50 supplemented with 10% FBS, and fixed at the indicated times. For rapamycin induction experiments, cells were treated with 100 nm or 500 nm rapamycin (Millipore, Cat #553210, Burlington, MA) for 1 hour followed by 2× PBS washout. To induce microtubule depolymerization, cells were treated with 10 μg/mL nocodazole (Sigma-Aldrich, Cat. #M1404, St. Louis, MO) or vehicle (dimethyl sulfoxide) for one hour at 37˚C. For microtubule regrowth experiments, cells were washed 3× with PBS after microtubule depolymerization, incubated in warm media for the indicated times, fixed, and stained. For HDAC inhibition, cells were treated with 2 μm tubacin (Cayman Chemicals, Cat #13691, Ann Arbor, MI) for the indicated times.

IMCD3 cells stably expressing GFP-PCM1-FKBP and HA-Kif5b-FRB were generated by co-transfecting cells with expression vectors followed by selection with 750 μg/mL G418 for 2 weeks and immunofluorescence-based screens for positive colonies. IMCD3 cells stably expressing mCherry-H2B were generated by infecting cells with mCherry-H2B–expressing lentivirus.

### Plasmids

Full-length cDNA of *Homo sapiens* PCM1 was amplified from peGFP-C1-PCM1 and cloned into peGFP-C1 (Clontech, Mountain View, CA) without stop codon using XhoI and KpnI sites. The cDNA of FKBP was amplified from PEX-RFP-FKBP by PCR and cloned into peGFP-PCM1 using KpnI and BamHI sites. PCM1-FKBP was amplified from the peGFP-C1-PCM1-FKBP by PCR and cloned into pcDNA3.1-myc-BirA* using XhoI and BamHI sites, pC4M-F2E-PEX-RFP-FKBP, PCI-neo-HA-BICD2-N (1–594)-FRB, pCI-neo-HI-Kif5b (1–807)-FRB, and pβactin-GFP-FRB-Kif17 (1–546) [22,23,48].

### Immunofluorescence and antibodies

Cells were grown on coverslips, washed twice with PBS, and fixed in either ice-cold methanol at −20˚C for 10 minutes or 4% PFA in cytoskeletal buffer (10 mM PIPES, 3 mM MgCl$_2$, 100 mM NaCl, 300 mM sucrose, pH 6.9) supplemented with 5 mM EGTA and 0.1% Triton X for 15 minutes at 37˚C. After PBS wash, cells were blocked with 3% BSA (Capricorn Scientific, Cat. #BSA-1T, Germany) in PBS + 0.1% Triton X-100 followed by incubation with primary

antibodies in blocking solution for 1 hour at room temperature. Cells were washed three times with PBS and incubated with secondary antibodies and DAPI (Thermo Scientific, Cat #D1306, Waltham, MA) at 1:2,000 for 45 minutes at room temperature. Following three washes with PBS, cells were mounted using Mowiol mounting medium containing N-propyl gallate (Sigma-Aldrich, St. Louis, MO). Primary antibodies used for immunofluorescence were mouse anti-acetylated tubulin (clone 6-11B, 32270, Thermo Fischer, Waltham, MA) at 1:10,000, mouse anti–gamma-tubulin (Sigma-Aldrich, clone GTU-88, T5326, St. Louis, MO) at 1:1,000, mouse anti-GFP (Thermo Scientific, A-11120, clone 3E6) 1:750, mouse anti–alpha-tubulin (Sigma-Aldrich, DM1A, St. Louis, MO) at 1:1,000, rabbit anti-CEP152 (Bethyl Laboratories, A302-480A, Montgomery, TX) at 1:500, rabbit anti-KIAA0753 (Sigma-Aldrich, HPA023494, St. Louis, MO) at 1:1,000, rabbit anti-CCDC14 (Genetex, GTX120754, Irvine, CA) at 1:1,000, rabbit anti-OFD1 (Proteintech, 22851-1-AP, Rosemont, IL) at 1:500, rabbit anti-Cep131 (Proteintech, 25757-1-AP, Rosemont, IL) at 1:1,000, mouse anti-centrin 3 (Abnova, H0007070-MO1, clone 3E6, Taiwan) at 1:500, rabbit anti-Cep290 (Abcam, ab84870, Cambridge, MA) at 1:1,000, anti-rabbit Cep72 (Bethyl Laboratories, A301-298A, Montgomery, TX) at 1:1,000, rabbit anti-Cep63 (Millipore, 61292 Burlington, MA) at 1:1,000, rabbit anti-MIB1 (Sigma-Aldrich, M5948, St. Louis, MO) at 1:1,000, mouse anti-C-NAP1 (Santa Cruz Biotechnology, sc 390540, Santa Cruz, CA) at 1:1,000, rabbit anti IFT88 (Proteintech, 13967-1-AP, Rosemont, IL) at 1:250, mouse GT335 (Adipogen, A27791601, San Diego, CA), rabbit Arl13B (Neuromab, Clone N295B/66, Davis, CA) at 1:500, and anti-Ninein (Abcam, ab4447, Cambridge, MA) at 1:500. Rabbit anti-PCM1, anti-Cep120, and anti-GFP antibodies were generated and used for immunofluorescence as previously described (Firat-Karalar et al., 2014). Secondary antibodies used for immunofluorescence experiments were Alexa Fluor 488-, 568-, or 633-coupled (Life Technologies, Carslbad, CA), and they were used at 1:2,000. Biotinylated proteins were detected with streptavidin coupled to Alexa Fluor 488 or 594 (1:1,000; Life Technologies, Carslbad, CA).

## Microscopy and image analysis

Time-lapse live imaging was performed with Leica SP8 confocal microscope equipped with an incubation chamber mounted on an automized stage. For cell cycle experiments, asynchronous cells were plated on LabTek dishes (Thermo Fisher, Waltham, MA) and imaged at 37°C with 5% $CO_2$ with a frequency of 4.5 minutes per frame, with 1-μm step size and 12-μm stack size in 512 × 512 pixel format at a specific position using an HC PL APO CS2 40× 1.3 NA oil objective. For time-lapse imaging experiments, HeLa cells were stained with SIR-Hoechst for visualization of chromosomes (100 nm, Spirochrome, Switzerland). For centrosomal protein level quantifications, images were acquired with Leica DMi8 inverted fluorescent microscope with a stack size of 10 μm and step size of 0.5 μm using an HC PL APO CS2 63× 1.4 NA oil objective. Higher resolution images were taken by using an HC PL APO CS2 63× 1.4 NA oil objective with Leica SP8 confocal microscope equipped with the SVI Huygens deconvolution suite.

Quantitative immunofluorescence for pericentrosomal levels of select proteins was performed by acquiring a z-stack of control and depleted cells using identical gain and exposure settings. The maximum-intensity projections were assembled from z-stacks. The centrosome region for each cell were defined by staining for a centrosomal marker gamma-tubulin. The region of interest that encompassed the centrosome was defined as a circle of 3-μm$^2$ area centered at the centrosome in each cell. Total pixel intensity of fluorescence within the region of interest was measured using ImageJ (National Institutes of Health, Bethesda, MD). Background subtraction was performed by quantifying the fluorescence intensity of a region of

equal dimensions in the area proximate to the centrosome. Statistical analysis was done by normalizing these values to their mean. Primary cilium formation was assessed by counting the total number of cells and the number of cells with primary cilia, as detected by DAPI and acetylated tubulin or Arl13b staining, respectively. Ciliary length was measured using acetylated tubulin or Arl13b as the ciliary length marker. All values were normalized relative to the mean of the control cells (= 1).

For functional assays, quantification and analysis were performed only in cells that exhibited complete redistribution of satellites to the cell periphery or center. Complete redistribution to the cell periphery was defined by lack of centrosomal GFP-PCM1-FKBP signal in the pericentrosomal area within a 3-$\mu m^2$ area centered on the centrosome. Complete redistribution to the cell center was defined by the lack of GFP-PCM1-FKBP signal in the cytoplasm beyond the 3 $\mu m^2$ pericentrosomal area. As controls, cells that were not treated with rapamycin in which satellites display their typical distribution pattern were quantified.

## Biotin-streptavidin affinity purification

BioID pulldown experiments were performed as previously described [49]. Briefly, HEK293T cells were transfected with myc-BirA*-PCM1-FKBP or myc-BirA*. Twenty-four hours post transfection, cells were incubated with DMEM-supplemented 10% FBS and 50 μM biotin for 18 hours. Cells were lysed in lysis buffer (50 mM Tris, pH 7.4, 500 mM NaCI, 0.4% SDS, 5 mM EDTA, 1 mM DTT, 2% Triton X-100, protease inhibitors), and soluble fractions were incubated overnight at 4˚C with streptavidin agarose beads (Thermo Scientific, Waltham, MA). Beads were washed twice with wash buffer 1 (2% SDS in dH2O), once with wash buffer 2 (0.2% deoxycholate, 1% Triton X-100, 500 mM NaCI, 1 mM EDTA, and 50 mM Hepes, pH 7.5), once with wash buffer 3 (250 mM LiCI, 0.5% NP-40, 0.5% deoxycholate, 1% Triton X-100, 500 mM NaCI, 1 mM EDTA, and 10 mM Tris, pH 8.1), and twice with wash buffer 4 (50 mM Tris, pH 7.4, and 50 mM NaCI). Ten percent of the sample was analyzed by western blotting to assess pulldown, and 90% of the sample was analyzed by mass spectrometry.

## Mass spectrometry and data analysis

After on-bead tryptic digest of biotinylated proteins, peptides were analyzed by online C18 nanoflow reversed-phase nano liquid chromatography (Dionex Ultimate 3000 RS LC, Thermo Scientific, Waltham, MA) combined with orbitrap mass spectrometer (Q Exactive Orbitrap, Thermo Scientific, Waltham, MA). Samples were separated in an in-house packed 75 μm i.d. × 23-cm C18 column (Reprosil-Gold C18, 3 μm, 200 Å, Dr. Maisch) using 75-minute linear gradients from 5%–25%, 25%–40%, 40%–95% acetonitrile in 0.1% formic acid with 300 nL/minute flow in 90 minutes of total run time. The scan sequence began with an MS1 spectrum (Orbitrap analysis; resolution 70,000; mass range 400–1,500 m/z; automatic gain control [AGC] target $1 \times 10^6$; maximum injection time, 32 ms). Up to 15 of the most intense ions per cycle were fragmented and analyzed in the orbitrap with data-dependent acquisition (DDA). MS2 analysis consisted of collision-induced dissociation (higher-energy collisional dissociation [HCD]) (resolution 17,500; AGC $1 \times 10^6$; normalized collision energy [NCE] 26; maximum injection time, 85 ms). The isolation window for MS/MS was 2.0 m/z. Raw files were processed with Thermo Proteome Discoverer 1.4.

Carbamidomethylation of cysteine was used as fixed modification, and acetylation (protein N termini) and oxidation of methionine were used as variable modifications. Maximal two missed cleavages were allowed for the tryptic peptides. The precursor mass tolerance was set to 10 ppm, and both peptide and protein false discovery rates (FDRs) were set to 0.01. The database search was performed against the human Uniprot database (release 2016).

Mass spectrometry data for each sample were derived from two biological and two technical replicates. Spectral counts of identified proteins were used to calculate the BFDR and probability of possible protein–protein interaction by SAINTexpress v.3.6.3 with -L option [50]. Proteins were filtered out according to SAINTscore ($>0.5$) and BFDR ($<0.05$). The cutoffs for enrichment ($>1.25$-fold relative to control) and depletion (0.75-fold relative to control) of proteins in cells with peripheral clusters were determined empirically by assessing which range in the comparative analysis include proteins we validated using immunofluorescence as part of peripheral clusters or previously implicated in cilium biogenesis. Interaction maps were drawn using Cytoscape. GO-enrichment analysis was done using EnrichR. The significantly altered GO categories (biological process and cellular compartment) were selected with a Bonferroni-adjusted cutoff $p$-value of 0.05.

## Statistical analysis

Statistical results and average and standard deviation values were computed and plotted by using Prism8 (GraphPad, La Jolla, CA). Two-tailed $t$ tests and one-way and two-way ANOVA tests were applied to compare the statistical significance of the measurements. Error bars reflect SD. The following key is used for asterisk placeholders for $p$-values in the figures: $^*p < 0.05$, $^{**}p < 0.01$, $^{***}p < 0.001$, $^{****}p < 0.0001$.

## Supporting information

**S1 Fig. Development and validation of the inducible trafficking assay. (A)** Localization of GFP-PCM1 and GFP-PCM1-FKBP in cells. HeLa cells were transfected with GFP-PCM1 or GFP-PCM1-FKBP, fixed after 24 hours, and stained for GFP, PCM1, and DAPI. **(B)** Effects of HA-BICD2-FRB or HA-Kif5b-FRB expression on satellite distribution. HeLa cells were transfected with HA-BICD2-FRB or HA-Kif5b-fRB, fixed after 24 hours, and stained for HA, PCM1, gamma-tubulin, and DAPI. **(C, D)** Validation of GFP-PCM1-FKBP and endogenous PCM1 mispositioning upon rapamycin-induced dimerization in cells. Hela cells co-expressing GFP-PCM1-FKBP with **(C)** HA-Kif5b-FRB or **(D)** HA-BICD2-FRB were treated with rapamycin for 1 hour, fixed 24 hours after transfection, and stained for GFP, HA, PCM1, and DAPI. Cells that were not treated with rapamycin were processed in parallel as controls. **(E)** Representation of partial distribution of satellites upon rapamycin induction. HeLa cells co-expressing GFP-PCM1-FKBP with HA-Kif5b-FRB or HA-BICD2-FRB were treated with rapamycin for 1 hour, fixed 24 hours after transfection, and stained for GFP, HA, PCM1, and DAPI. Partial distribution was defined by GFP-PCM1-FKBP signal in the pericentrosomal area in Kif5b-expressing cells and signal in the region excluding the centrosomal area in BICD2-expressing cells. **(F)** Expression of GFP-PCM1-FKBP with HA-Kif5b-FRB or HA-BICD2-FRB and their redistribution upon rapamycin induction do not perturb the microtubule network. Cells were stained for GFP, alpha-tubulin, and DAPI. **(G)** Rapamycin treatment did not perturb satellite distribution in wild-type cells and cells expressing only GFP-PCM1-FKBP. Cells were treated with rapamycin for 1 hour, fixed after 24 hours, and stained for GFP or PCM1, gamma-tubulin, and DAPI. **(H)** Co-expression of GFP-PCM1-FKBP with the constitutively active HA-Kif17 (1–181 aa)-FRB targets satellites to the cell periphery, where satellite clusters are heterogeneously distributed. Transfected HeLa cells were treated with rapamycin for 1 hour, fixed after 24 hours, and stained for GFP, PCM1, gamma-tubulin, and DAPI. Scale bars, 10 μm; all insets show 4× enlarged centrosomes. BICD2, bicaudal D homolog 2; FKBP, FK506 binding protein 12; FRB, FKBP12-rapamycin-binding; GFP, green fluorescent protein; HA, hemagglutinin; Kif5b, kinesin family member 5b; PCM1, pericentriolar material 1 (TIF)

**S2 Fig. Effects of satellite mispositioning on the pericentrosomal levels of various satellite residents. (A)** HeLa cells co-expressing GFP-PCM1-FKBP with HA-Kif5b-FRB or HA-BICD2-FRB were treated with rapamycin for 1 hour followed by fixation at 6 and 24 hours. Cells that were not treated with rapamycin and exhibited pericentrosomal clustering of GFP-PCM1-FKBP–like endogenous PCM1 of wild-type cells were processed in parallel with controls. Cells were stained with antibodies anti-GFP to identify cells with complete redistribution to the cell periphery or center, anti–gamma-tubulin to mark the centrosome, and antibodies against the indicated proteins. Fluorescence intensity at the centrosome was quantified and average means of the levels in control cells were normalized to 1. $n \geq 25$ cells per experiment. Data represent the mean value from two experiments per condition ± SD (**$p < 0.01$, ***$p < 0.001$, ****$p < 0.0001$, n.s. nonsignificant). Error bars = SD. Source data can be found in S3 Data. **(B)** Control and rapamycin-treated cells were stained for GFP, gamma-tubulin, and indicated satellite proteins. Images represent centrosomes in cells from the same coverslip taken with the same camera settings. DNA was stained with DAPI. Cell edges are outlined. Scale bars, 10 µm; all insets show 4× enlarged centrosomes. BICD2, bicaudal D homolog 2; FKBP, FK506 binding protein 12; FRB, FKBP12-rapamycin-binding; GFP, green fluorescent protein; HA, hemagglutinin; Kif5b, kinesin family member 5b; PCM1, pericentriolar material 1
(TIF)

**S3 Fig. Effects of satellite misdistribution on microtubule nucleation and daughter centriole composition. (A)** The daughter centriole protein Cep120 was redistributed to the mother centriole in BICD2-expresing cells with centrosomal satellite accumulation. HeLa cells co-expressing GFP-PCM1-FKBP with HA-BICD2-FRB were treated with rapamycin for 1 hour, fixed at 24 hours, and stained for GFP, Cep120, Cep164, and DAPI. Cells that were not treated with rapamycin were used as a control. **(B)** Gamma-tubulin localization in control cells and in Kif5b-expressing cells with peripheral satellite clustering. HeLa cells co-expressing GFP-PCM1-FKBP with HA-Kif5b-FRB were treated with rapamycin for 1 hour, fixed at 24 hours, and stained for GFP, gamma-tubulin, and DAPI. Images represent centrosomes in cells from the same coverslip taken with the same camera settings. Cells that were not treated with rapamycin were processed in parallel as a control. Fluorescence intensity at the centrosome was quantified, and average mean of the levels in control cells were normalized to 1. $n \geq 25$ cells per experiment. Data represent mean value from two experiments per condition ± SD (n.s. nonsignificant, ****$p < 0.0001$). Error bars = SD. Source data can be found in S3 Data. **(C)** Effect of gamma-tubulin accumulation at the peripheral satellites on microtubule nucleation. Rapamycin-treated IMCD3$^{peripheral}$ cells were treated with DMSO or 10 µg/mL nocodazole for 1 hour. After microtubule depolymerization, cells were washed, incubated with complete media for the indicated times, fixed, and stained for GFP, alpha-tubulin, and DAPI. **(C)** Rapamycin-treated IMCD3$^{peripheral}$ cells were treated with DMSO or 10 µg/mL nocodazole for 1 hour. After microtubule depolymerization, cells were washed, fixed 10 minutes after nocodozole washout, and stained for GFP, alpha-tubulin, ninein, and DAPI. Scale bars, 10 µm; all insets show 3× enlarged centrosomes. BICD2, bicaudal D homolog 2; FKBP, FK506 binding protein 12; FRB, FKBP12-rapamycin-binding; GFP, green fluorescent protein; HA, hemagglutinin; Kif5b, kinesin family member 5b; PCM1, pericentriolar material 1
(TIF)

**S4 Fig. Comparative analysis of the PCM1 proximity interactome of control cells and cells with peripheral satellite targeting. (A)** The control and peripheral PCM1 interactomes share 476 components. A total of 65 proteins were specific to peripheral satellites and 125 proteins

were specific to control cells. **(B)** GO-enrichment analysis of the proteins enriched (>1.25-fold relative to control) and depleted (<0.75-fold relative to control) in cells with peripheral satellite clustering after rapamycin treatment based on their biological processes. The x-axis represents the log-transformed $p$-value (Fisher's exact test) of GO terms. Source data can be found in S4 Data. GO, Gene Ontology; PCM1, pericentriolar material 1
(TIF)

**S5 Fig. Effects of rapamycin treatment and satellite peripheral localization on cilium assembly, maintenance, and disassembly. (A)** Validation of IMCD3$^{peripheral}$ cells for expression of HA-Kif5b and GFP-PCM1-FKBP. Control and rapamycin-induced IMCD3$^{peripheral}$ cells were stained for GFP, HA, gamma-tubulin, and DAPI. **(B)** GFP-PCM1-FKBP restores ciliogenesis defects of satellite-less IMCD3 PCM1 KO cells and peripheral satellite clustering compromises their ciliogenesis efficiency. IMCD3 PCM1 KO$^{peripheral}$ cells stably expressing GFP-PCM1-FKBP and HA-Kif5b-FRB (−Rapa and +Rapa) and IMCD3 PCM1 KO cells were serum-starved for 48 hours, and the percentage of ciliated cells was determined by staining for acetylated-tubulin. Results shown are the mean of three independent experiments ± SD (= 50 cells/experiment, $^{*}p < 0.05$, $^{**}p < 0.01$). **(C)** Quantification of the percentage of ciliogenesis in control and rapamycin-treated IMCD3$^{peripheral}$ cells 48 hours after serum starvation. Results shown are the mean of two independent experiments ± SD (= 50 cells/experiment, $^{**}p < 0.01$). **(D)** Rapamycin treatment by itself in IMCD3 cells does not affect the efficiency of ciliogenesis and cilium maintenance. The experiments were performed in IMCD3 cells following the experimental outline for ciliogenesis and maintenance experiments in Fig 4, and the fraction of ciliated cells were quantified after 48 hours of serum starvation for the cilium assembly experiment and 24 hours after rapamycin treatment for maintenance experiments. Results shown are the mean of two independent experiments ± SD (= 200 cells/experiment). **(E)** Effect of peripheral satellite clustering on cilium maintenance. IMCD3$^{peripheral}$ cells were serum-starved for 48 hours, treated with rapamycin for 1 hour, and the percentage of ciliated cells was determined over 24 hours by staining for acetylated tubulin. Cells that were not treated with rapamycin were used as a control. Data represent the mean value from two experiments per condition ± SD ($^{***}p < 0.001$, $^{****}p < 0.0001$). **(F)** Effect of peripheral satellite clustering on cilium disassembly. IMCD3$^{peripheral}$ were serum-starved for 48 hours, treated with rapamycin for 1 hour, induced by serum stimulation, and the percentage of ciliated cells was determined over 6 hours by staining for acetylated tubulin. Cells that were not treated with rapamycin were used as a control. Data represent mean value from two experiments per condition ± SD ($^{*}p < 0.1$). Error bars = SD. Source data can be found in S5 Data. FKBP, FK506 binding protein 12; FRB, FKBP12-rapamycin-binding; GFP, green fluorescent protein; HA, hemagglutinin; IMCD3$^{peripheral}$, inner medullary collecting duct; Kif5b, kinesin family member 5b; KO, knockout; n.s., nonsignificant; PCM1, pericentriolar material 1; Rapa, rapamycin
(TIF)

**S6 Fig. Phenotypic consequences of satellite redistribution to the cell center or cell periphery on mitotic progression and time. (A)** Rapamycin treatment itself does not affect mitotic time and cytokinesis time of IMCD3 PCM1 KO cells. Control and rapamycin-treated IMCD3 PCM1 KO cells were imaged over 16 hours using time-lapse imaging. Mitotic and cytokinesis times were quantified. Data represent mean value from two experiments per condition ± SD. **(B)** Effect of peripheral and centrosomal satellite clustering on mitotic dynamics of satellites and mitotic progression of HeLa cells. GFP-PCM1-FKBP was co-transfected to HeLa cells with HA-Kif5b-FRB or HA-BICD2-FRB, treated with rapamycin, and imaged every 4.5 minutes for 16 hours. Representative brightfield and GFP-PCM1-FKBP fluorescence still frames

from time-lapse experiments for the different phenotypes quantified were shown. Cell edges are outlined. Scale bar, 10 μm. **(C, D)** Quantification of (B). Percentage of control, and rapamycin-treated mitotic cells were imaged by the time-lapse imaging and videos were analyzed to classify mitotic cells into categories of cells that completed mitosis, cells that underwent apoptosis, and cells arrested in mitosis for more than 5 hours. $n > 20$ cells per experiment. Data represent mean ± SD of three independent experiments. **(E)** Effect of peripheral and centrosomal satellite clustering on mitotic time in HeLa cells co-expressing GFP-PCM1-FKBP with HA-Kif5b-FRB or HA-BICD2-FRAB. Control and rapamycin-treated cells were stained with SIR-DNA and imaged by time-lapse microscopy. **(F)** Quantification of (E). $n > 20$ mitotic cells per experiment. Data represent mean ± SD of two independent experiments. Error bars = SD. Source data can be found in S6 Data. BICD2, bicaudal D homolog 2; FKBP, FK506 binding protein 12; FRB, FKBP12-rapamycin-binding; GFP, green fluorescent protein; HA, hemagglutinin; IMCD3, inner medullary collecting duct; Kif5b, kinesin family member 5b; KO, knockout; n.s., non-significant; PCM1, pericentriolar material 1; SIR, silicone rhodamine (TIF)

**S1 Table. Raw mass spectrometry data for Myc-BirA\*-PCM1-FKBP with and without rapamycin treatment in Kif5b-expressing cells.** Mass spectrometry analysis of proximity interactors of Myc-BirA\*-PCM1-FKBP proteins. Proximity interactors from two experimental and two technical replicates for each condition were analyzed by SAINT. FKBP, FK506 binding protein 12; Kif5b, kinesin family member 5b; PCM1, pericentriolar material 1; SAINT, significance analysis of interactome (XLSX)

**S2 Table. Unique proteins identified in control and rapamycin-treated Myc-BirA\*-PCM1-FKBP.** Proteins with BFDR ($<0.01$) values were defined as part of the PCM1 interactomes. **Tab 1:** Unique peptides identified in control and rapamycin-treated cells (−Rapa versus +Rapa)**. Tab 2:** Proteins enriched ($>1.25$-fold relative to control), depleted ($<0.75$-fold relative to control), and unaltered in rapamycin-treated cells relative to control cells. BFDR, Bayesian false discovery rate; FKBP, FK506 binding protein 12; PCM1, pericentriolar material 1; Rapa, rapamycin (XLSX)

**S3 Table. GO-enrichment analysis of the PCM1 interactors at the cell periphery.** Proximity interactors of Myc-BirA\*-PCM1-FKBP were classified into GO categories using the EnrichR analysis program. FKBP, FK506 binding protein 12; GO, Gene Ontology; PCM1, pericentriolar material 1 (XLSX)

**S1 Movie. Live imaging of HeLa cells co-expressing GFP-PCM1-FKBP and HA-Kif5b before and after rapamycin treatment.** FKBP, FK506 binding protein 12; GFP, green fluorescent protein; HA, hemagglutinin; Kif5b, kinesin family member 5b; PCM1, pericentriolar material 1 (AVI)

**S2 Movie. Live imaging of HeLa cells co-expressing GFP-PCM1-FKBP and HA-Kif5b before and after rapamycin treatment.** FKBP, FK506 binding protein 12; GFP, green fluorescent protein; HA, hemagglutinin; Kif5b, kinesin family member 5b; PCM1, pericentriolar material 1 (AVI)

**S3 Movie. Live imaging of rapamycin-induced HeLa cells co-expressing GFP-PCM1-FKBP and HA-Kif5b after nocodazole treatment.** FKBP, FK506 binding protein 12; GFP, green fluorescent protein; HA, hemagglutinin; Kif5b, kinesin family member 5b; PCM1, pericentriolar material 1
(AVI)

**S4 Movie. Live imaging of rapamycin-induced HeLa cells co-expressing GFP-PCM1-FKBP and HA-BICD2 after nocodazole treatment.** BICD2, bicaudal D homolog 2; FKBP, FK506 binding protein 12; GFP, green fluorescent protein; HA, kinesin family member 5b; PCM1, pericentriolar material 1
(AVI)

**S1 Data. Quantification of pericentrosomal PCM1 levels before and after rapamycin induction of satellite mispositioning.** The data and statistical analysis that correspond to Fig 1E and 1F. PCM1, pericentriolar material 1
(XLSX)

**S2 Data. Quantification of pericentrosomal levels of indicated proteins before and after rapamycin induction of satellite mispositioning.** The data and statistical analysis that correspond to Fig 2C.
(XLSX)

**S3 Data. Quantification of pericentrosomal levels of indicated proteins before and after rapamycin induction of satellite mispositioning.** The data and statistical analysis that correspond to S2A Fig and S3B Fig.
(XLSX)

**S4 Data. Quantification of GO categories enriched in PCM1 proximity maps.** The data and statistical analysis that correspond to Fig 3C and 3D and S4B Fig. GO, Gene Ontology; PCM1, pericentriolar material 1
(XLSX)

**S5 Data. Quantification of cilium assembly, maintenance, and disassembly phenotypes before and after rapamycin induction of satellite mispositioning.** The data and statistical analysis that correspond to Fig 4C–4F and S5B–S5F Fig.
(XLSX)

**S6 Data. Quantification of mitotic progression and mitotic time in IMCD3 and HeLa cells before and after rapamycin induction of satellite mispositioning.** The data and statistical analysis that correspond to Fig 5C and 5D and S6A, 6C, S6D and S6F Fig. IMCD3, inner medullary collecting duct
(XLSX)

**S1 Raw Images. Western blot confirmation of pulldown of biotinylated proteins and myc-BirA\*-PCM1-FKBP in BioID experiments.** The unprocessed western blot data for Fig 3B. BioID, Biotin Identification; FKBP, FK506 binding protein 12; PCM1, pericentriolar material 1
(PDF)

## Acknowledgments

We acknowledge all members of Firat-Karalar laboratory and Jennifer Wang from the Stearns laboratory at Stanford University for insightful discussions regarding this work. FKBP and

FRB constructs were a kind gift from Lukas Kapitein (Utrecht University). We acknowledge the Koç University Proteomics Facility for mass spectrometry analysis, Altug Kamacioglu for the SAINT analysis of the mass spectrometry data, and Mudasir Banday for cloning support. We acknowledge Life Science Editors for manuscript editing assistance. We acknowledge the members of the University of Washington Cilia Journal club for providing feedback on the bioRxiv version of our manuscript.

## Author Contributions

**Conceptualization:** Özge Z. Aydin, Sevket Onur Taflan, Elif Nur Firat-Karalar.

**Data curation:** Özge Z. Aydin, Sevket Onur Taflan, Elif Nur Firat-Karalar.

**Formal analysis:** Özge Z. Aydin, Sevket Onur Taflan, Elif Nur Firat-Karalar.

**Funding acquisition:** Elif Nur Firat-Karalar.

**Investigation:** Özge Z. Aydin, Sevket Onur Taflan, Elif Nur Firat-Karalar.

**Methodology:** Özge Z. Aydin, Sevket Onur Taflan, Can Gurkaslar, Elif Nur Firat-Karalar.

**Project administration:** Elif Nur Firat-Karalar.

**Resources:** Elif Nur Firat-Karalar.

**Supervision:** Özge Z. Aydin, Elif Nur Firat-Karalar.

**Validation:** Özge Z. Aydin, Sevket Onur Taflan, Elif Nur Firat-Karalar.

**Visualization:** Özge Z. Aydin, Sevket Onur Taflan, Elif Nur Firat-Karalar.

**Writing – original draft:** Elif Nur Firat-Karalar.

**Writing – review & editing:** Özge Z. Aydin, Sevket Onur Taflan, Elif Nur Firat-Karalar.

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
