## [Editor Report · Decision Letter 0]

3 Feb 2020

Dear Dr Firat-Karalar, 

Thank you for submitting your manuscript entitled "Acute inhibition of centriolar satellite function and positioning reveals their functions at the primary cilium" for consideration as a Short Reports by PLOS Biology.

Your manuscript has now been evaluated by the PLOS Biology editorial staff as well as by an academic editor with relevant expertise and I am writing to let you know that we would like to send your submission out for external peer review.

Please re-submit your manuscript within two working days, i.e. by Feb 05 2020 11:59PM.

Kind regards,

Lauren A Richardson, Ph.D

Senior Editor

PLOS Biology

---

## [Decision Letter · Decision Letter 1]

13 Mar 2020

Dear Dr Firat-Karalar,

Thank you very much for submitting your manuscript "Acute inhibition of centriolar satellite function and positioning reveals their functions at the primary cilium" for consideration as a Short Report at PLOS Biology. Your manuscript has been evaluated by the PLOS Biology editors, an Academic Editor with relevant expertise, and by three independent reviewers.

As you will see, the reviewers all seem very positive and find your conclusions interesting and significant for the field. However, they also recommend that you perform some experiments to strengthen the conclusions and also ask you to clarify several points.

In light of the reviews (attached below), we will not be able to accept the current version of the manuscript, but we would welcome re-submission of a much-revised version that takes into account the reviewers' comments. We cannot make any decision about publication until we have seen the revised manuscript and your response to the reviewers' comments. Your revised manuscript is also likely to be sent for further evaluation by the reviewers.

We expect to receive your revised manuscript within 2 months. 

**IMPORTANT - SUBMITTING YOUR REVISION**

*Re-submission Checklist*

*Published Peer Review*

*PLOS Data Policy*

*Blot and Gel Data Policy*

Sincerely,

Ines

--

Ines Alvarez-Garcia, PhD

Senior Editor

PLOS Biology

Carlyle House, Carlyle Road

Cambridge, CB4 3DN

+44 1223–442810

Reviewers’ comments

Rev. 1:

The manuscript by Aydin et al. deals with the fascinating and contentious issue of the function of centriolar satellites. They developed a clever rapamycin inducible approach to translocate satellites either to the cell periphery or to the centrosome. Using these tools they perform BioID mass spec analysis to try and determine the components of the satellites away from the potentially contaminating centrosome. Using this approach they found many of the known satellite proteins but also identified some that may warrant a deeper look based on the differential enrichment. Finally and most significantly they utilize these tools to address a more nuanced function for satellites and identify 2 novel functions in cilia maintenance and disassembly. I think this is an interesting paper that has identified some important and novel functions for satellites. However there a couple of experiments that could potentially really strengthen the paper and are worth considering.

Comments:

My primary suggestion to improve this paper would be to perform the functional experiments with the BICD2-PCM1 constructs. The authors state "In contrast to HA-Kif5b, we noticed that HA-BICD2 expression by itself resulted in satellite dispersal throughout the cytosol (Fig. S1B), which is consistent with its previous characterization as a dominant negative mutant that impairs dynein-dynactin function [24-26]. Therefore, we did not use BICD2-based centrosomal satellite clustering as a way to assay satellite functions in ciliated and proliferating cells." However, I feel that an opportunity has been missed. While I agree that the DN dispersal phenotype complicates things, I still think that there is something to be gained from the forced enrichment of PCM1 at the centrosome (which is considerable). Does this enrichment disperse in a cell cycle dependent manner (e.g. Fig 5A) or is it forced to maintain its centrosomal localization and if so to what end. Similarly, one could predict that this enrichment might enhance ciliogenesis or maintenance above WT levels or slow disassembly. This analysis feels lacking from the data.

In Figure S3C it is hard to make the firm conclusion that the ectopic satellites are not acting as an MTOC. This is due to the large amount of MTs after 15min, I think doing it at 5min would provide more useful information. Or perhaps showing the lack of a MT - end protein such as ninein.

Minor:

The description of Cep164 localization as a ring like localization seems misleading. I suspect that it is likely spherical but that given a the imaging plane looks ring like.

"Cep120 was strongly enriched in the mother centriole in BICD2-expressing cells (Fig. 3A)." Should be Fig. S3A.

On Page 10 Figure 3D should be 3E.

Error bars on Figure S5A

I found these sentences confusing: "While 70.9% ± 3.3 of control cells were ciliated, only 52.8% ± 0.9 of IMCD3 PCM1 KOperipheral cells were ciliated (p<0.01) (Fig. 4C). Similarly, only 77.35% ± 1.65% IMCDperipheral cells ciliated relative to 54.4% ± 1.4 control ciliating population (p<0.01 ) (Fig. 4D)." please clarify.

The figures need to be cleaned up with proper scale bars and such. 5A cells do not seem to be at the same scale.

The movies were not properly referenced in the text.

Rev. 2: Philippe Bastin – please note that this reviewer has waived anonymity

This manuscript addresses the role of the cellular positioning of centrosome satellites in their function. For this, the authors use an elegant approach to conditionally reposition the central satellite protein PCM1 to the cell periphery (use of kinesin) or the cell centre (use of dynein). After a clear and well-documented description of the system, two major findings came up:

(1) Some centrosomal proteins are "trapped" with PCM1 and have limited/no access to the centrosome while others are not affected, indicated two possible pathways for centrosomal targeting. This is supported by immunofluorescence assays and proteomic studies with proximity labelling.

(2) The system is used to investigate convincingly the contribution of PCM1 and centriolar satellites in the maintenance and the disassembly of the primary cilium whereas inhibition of satellite dispersion during mitosis does not seem to cause mitotic defects

Overall, it is a nice piece of work that is well executed and well written. The authors also discuss the limitations of their system (for example satellite dispersion upon HA-BICD2 expression). The message of the paper will be of interest for cell biologists interested in organelle function and positioning, an important theme in developmental and cellular biology. One might say that the results were somehow expected since structures matter where they are but this reviewer believes in data and getting experimental evidence is therefore essential. The data reported are quite compelling, although they could be further discussed. I have several suggestions that I hope will contribute to improve the manuscript. Most things can be addressed without further experiments (or by including data that are possibly available), although some experiments could bring useful information (see below). In my view, reviewing is a dialogue and I will be happy to read the response of the authors.

1. The discussion is a bit superficial and focuses a lot on future work, it somehow looks more similar to a grant application. This reviewer believes that the authors have nice and exciting data that deserve more exhaustive discussion. Here are a few thoughts: centrosomal proteins seem to fall in two categories: those that rely on PCM1 and those that don't. What is the relationship between proteins belonging to each of this group? Are they known to interact? What about the conservation profile? PCM1 is not present in all eukaryotic species. Does its presence somehow correlate with the presence of proteins that rely on satellite for centrosomal targeting? Is there a relationship with proteins associated to cilia or only found at the centrosome level?

2. I was a bit surprised to see that cilium length was not reported (although the use of ARL13b/acetylated tubulin for measurements is mentioned in M&M). Possibly the authors have the data. The reason why it is not mentioned might be found in a recent publication of the same group (Odabasi et al EMBO19) that showed that deletion of PCM1 reduces the number of cilia but not the length of remaining cilia in IMCD3 cells. In any case, the length of the remaining cilia should be discussed here. It is possible that the phenotype would be more pronounced if it was taken into account. I agree it sounds unlikely given the KO results but as said above, evidence is better. Nevertheless, it would not be the first time that a mutant/modification of a protein (in this case its location) induces a stronger phenotype than a deletion.

If cilium length is unaffected, that means we are in an "all or nothing" situation. Either cells manage to do a cilium of normal length, either they don't and in that case. How do the authors interpret this? This is worth discussing.

3. This reviewer likes the proteomic data but is a bit amazed that some of the results are not experimentally validated/investigated a bit further. For example, the presence of many members of the intraflagellar transport B (IFTB) complex is remarkable and could suggest a depletion of these factors essential for cilium elongation. This could help answering the question raised above. In case enough IFTB proteins are available, the cilium might elongate. If not, IFT trafficking might be insufficient to sustain ciliogenesis initiation. This could be relevant in case the cilium forms like in Chlamydomonas where most of the IFT material is injected at once at the onset of ciliogenesis (see papers of Wallace Marshall MBoC2005; JCB2009). I could not find the information for IMCD cilium assembly though. Immuofluorescence with anti-IFT antibodies in conditions where PCM1 and satellites are sent to the periphery could be very useful in proving (or not) this hypothesis. I know that the quality of anti-IFT antibodies in mammalian cells has been an issue for the field, but I will be happy to hear the opinion of the authors.

4. Finally, the authors did an excellent job for the investigation of cilium maintenance and disassembly with several data points (Fig. 4E&F). By contrast, only one time point is shown for the assembly (48h, Fig. 4B). At that stage, cilia are likely to have reached mature length. This is unfortunate because following how cilia grow with intermediate time points would give really nice insights on the way they rely on PCM1 proper location for construction. I don't know if the authors have done this experiment already (it's possibly another story!) but I would advise them to have a try.

Philippe Bastin, Institut Pasteur, Paris.

Rev. 3:

This manuscript documents the development of methods based on rapamycin-induced heterodimerization of FKBP and FRB to redistribute pericentriolar satellites from their normal positions, loosely dispersed around the centrosome, either to clusters at the periphery of the cell (minus the centrosome) or to a single large cluster focused on the mother centriole. The methods are validated in great detail. The authors then use immunofluorescence microscopy to analyze how select satellite proteins change in abundance when the satellites are repositioned. Using BioID, they next determine that the satellite interactome remains relatively unaltered when the satellites are redistributed to peripheral clusters. The authors then carry out studies to determine how ciliary assembly, maintenance, and disassembly are affected by repositioning of the satellites to the cell periphery. The results provide the first direct evidence that proximity of the satellites to the cilium’s base is important for primary cilium maintenance and disassembly. In addition to extending the general importance of cell organelle positioning for function of the cell organelle, the results raise interesting questions for future research. In general, the manuscript is scholarly, well written, and scientifically sound; it is appropriate for publication in PLOS Biology. However, it would be improved if the following issues were addressed prior to publication.

P. 4: The major section that begins here is called “Results and Discussion.” There also is a “Discussion” section that begins on p. 14. The section beginning on p. 4 probably should be simply “Results.”

P. 5: The phrase “co-expressing the FKBP-fusion of satellites as the cargo of interest” is awkward and unclear.

P. 5: HA-BICD2 expression by itself caused satellite dispersal, presumably as a result of its previously documented dominant-negative effect on dynein-dynactin function. Here or later, briefly explain why addition of rapamycin to these cells nevertheless results in clustering of satellites at the centrosome, presumably due to dynein moving on microtubules to their minus ends.

P. 7: Clustering of satellites is dependent on microtubules, but the clusters do not disperse when microtubules are depolymerized. Briefly discuss the likely reason that the clusters remain intact in the absence of microtubules.

P. 8: “except for centrin.” I could not find data for centrin. Add “data not shown”?

P. 9: “Despite prominent microtubule nucleation at the centrosomes, microtubule nucleation was not initiated at the peripheral gama-tubulin-positive clusters.” This conclusion is not supported by Fig. S3C. In the figure, it appears that there may be microtubule nucleation proportional to the amount of gamma-tubulin in the peripheral cluster as compared to the centrosome. Please provide clearer example or rephrase conclusion. In same sentence, change “gama-tubulin” to “gamma-tubulin.” In same figure panel, place numbers over inset columns corresponding to boxes in large image.

P. 10. The authors discuss “proteins enriched (>1.25 fold relative to control) or depleted (<0.75 fold relative to control).” Cutoffs of >1.25 fold and <0.75 fold do not seem very stringent for concluding that proteins are enriched or depleted. Please explain why these cutoffs were chosen and why such small changes are meaningful.

P. 10: “To gain insight into the specific ciliary processes that satellites regulate, we determined the PCM1 proximity interactors associated with the primary cilium in previous studies and classified them based on their functions in the ciliogenesis program in an interaction network (Fig. 3D).” Please provide citations for “previous studies.” “(Fig. 3D)” here should be “(Fig. 3E).”

P. 11: “Expression of GFP-PCM1-FKBP restored the ciliogenesis defect of IMCD3 PCM1 KO cells, confirming that this fusion protein is fully functional (Fig. S5A).” Without error bars on Fig S5A, it is not clear that this conclusion is supported by the data. Please add error bars or revise statement.

P. 19: “The region of interest that encompassed the centrosome was defined as a circle 3 μm2 area centered at the centrosome in each cell.” This and similar definitions currently in the Materials and Methods section are sufficiently important for understanding the experiments that they probably should be in the relevant places in the Results section.

P. 24: “Images represent centrosomes in cells from the same coverslip taken with the same camera settings.” As written, I am not sure this statement is needed or even relevant to the images shown. It presumably refers to the data in the scatterplots.

P. 29: “Figure S5. Rapamycin treatment does not interfere with ciliogenesis, cilium maintenance and cilium disassembly.” The title of the legend is confusing as Fig. S5D does show an experiment in which rapamycin treatment (and peripheral clustering) has an effect on ciliary maintenance. Perhaps rephrase title of figure legend.

The authors acknowledge Life Science Editors for editing assistance. The writing is indeed well edited up to about p. 9, and then becomes more error prone until the Discussion. Minor errors or omissions:

P. 8: 6th line from bottom: “(Fig. 3A)” should be “Fig. S3A.”

P. 9, “. . . how the composition of satellites upon their peripheral targeting . . .” Insert “changed” after “satellites”?

P. 10, line 8: I could not find a “Table 2.” This probably should be “Table S2.”

P. 10, last sentence: “composition peripheral satellites.” Insert “of” after “composition”?

P. 11: “we used IMCD3peripheral experiments for subsequent experiments at the cilia.” Replace first “experiments” with “cells”? Replace “at” with “on”?

P. 14: Decimal place is indicated by period two places (31.4; 32.8) but by comma three places (32,95; 48,56; 12,94).

P. 15: “suggesting that satellites functions upstream of HDAC6.” Replace “functions” with “function.”

P. 20: “After on-based tryptic digest.” What is “on-based”?

P. 27, Figure S1 legend: “excluding therosomal area in BICD2-expressing cells.” “therosomal”??

P. 27, Figure S1 legend: “F) Coexpression of GFP-PCM1-FKBP with the constitutively active HA-Kif17.” “F” should be “H.”

Figure S2B: Add “gamma” to “-tubulin” at top left.

---

## [Editor Report · Decision Letter 2]

27 Apr 2020

Dear Elif,

Thank you for submitting your revised Short Report entitled "Acute inhibition of centriolar satellite function and positioning reveals their functions at the primary cilium" for publication in PLOS Biology. I have now discussed your revision with the other editors and obtained advice from the Academic Editor. 

We're delighted to let you know that we're now editorially satisfied with your manuscript. However before we can formally accept your paper and consider it "in press", we also need to ensure that your article conforms to our guidelines. A member of our team will be in touch shortly with a set of requests. As we can't proceed until these requirements are met, your swift response will help prevent delays to publication. Please also make sure to address the data and other policy-related requests noted at the end of this email.

*Copyediting*

*Published Peer Review History*

*Early Version*

*Submitting Your Revision*

Best wishes,

Ines

--

Ines Alvarez-Garcia, PhD

Senior Editor

PLOS Biology

Carlyle House, Carlyle Road

Cambridge, CB4 3DN

+44 1223–442810

DATA POLICY:

Fig. 1E, F; Fig. 2C; Fig. 3C, D; Fig. 4C, D, E, F; Fig. 5C, D; Fig. S2A; Fig. S3B; Fig. S4B; Fig. S5B, C, D, E, F and Fig. S6A, C, D, F

---

## [Editor Report · Decision Letter 3]

4 Jun 2020

Dear Dr Firat-Karalar,

On behalf of my colleagues and the Academic Editor, Renata Basto, I am pleased to inform you that we will be delighted to publish your Short Reports in PLOS Biology. 

Early Version

PRESS 

Kind regards,

Alice Musson

Publishing Editor, 

PLOS Biology

on behalf of

Ines Alvarez-Garcia,

Senior Editor

PLOS Biology